ecology, health and disease and epidemiology

force of infection, host specificity, multi-host system, SIR model, transmission dynamics

**Author for correspondence:**
Skylar R. Hopkins
e-mail: skylar_hopkins@ncsu.edu

# Host preferences inhibit transmission from potential superspreader host species

Skylar R. Hopkins[1,2,3], Cari M. McGregor[2], Lisa K. Belden[1] and Jeremy M. Wojdak[2]

[1]Department of Applied Ecology, North Carolina State University, Raleigh, NC, USA
[2]Department of Biological Sciences, Virginia Tech, Blacksburg, VA, USA
[3]Department of Biology, Radford University, Radford, VA, USA

SRH, 0000-0002-8381-0601

Host species that are particularly abundant, infectious and/or infected tend to contribute disproportionately to symbiont (parasite or mutualist) maintenance in multi-host systems. Therefore, in a facultative multi-host system where two host species had high densities, high symbiont infestation intensities and high infestation prevalence, we expected interspecific transmission rates to be high. Instead, we found that interspecific symbiont transmission rates to caged sentinel hosts were an order of magnitude lower than intraspecific transmission rates in the wild. Using laboratory experiments to decompose transmission rates, we found that opportunities for interspecific transmission were frequent, where interspecific and intraspecific contact rate functions were statistically indistinguishable. However, most interspecific contacts did not lead to transmission events owing to a previously unrecognized transmission barrier: strong host preferences. During laboratory choice experiments, the symbiont preferred staying on or dispersing to its current host species, even though the oligochaete symbiont is a globally distributed host generalist that can survive and reproduce on many snail host species. These surprising results suggest that when managing symbiont transmission, identifying key host species is still important, but it may be equally important to identify and manage transmission barriers that keep potential superspreader host species in check.

## 1. Introduction

Most parasites, and symbionts more broadly, are maintained in multi-host communities by one or several key host species [1–6]. At one extreme, a single reservoir or maintenance host species maintains a symbiont in an apparent multi-host community [3,7–9], where the symbiont cannot persist without that one key host species. At the other extreme, true or obligate multi-host symbionts cannot persist without multiple host species [7,9], including symbionts with complex life cycles that must sequentially infect intermediate and definitive hosts. Between these extremes, there are facultative multi-host symbionts, which do not need multiple host species, but which can be independently maintained by any one of several maintenance host species [7,9]. Given this complexity, identifying which host species contribute the most to transmission and population maintenance remains challenging, especially in emerging disease systems [9–12]. However, if key host species can be identified and targeted, efforts to manage symbiont transmission can be more efficient and effective [7,8,11].

Three transmission-amplifying characteristics have been proposed for key host species [9] (figure 1): super abundant host species have high densities and are thus likely to have high density-dependent contact rates [9,13]; super infectious host species have high transmission potential during contacts, such as when they have high pathogen loads and thus deliver a high dose [9,14,15]; and super infected host species have high infection prevalence, such

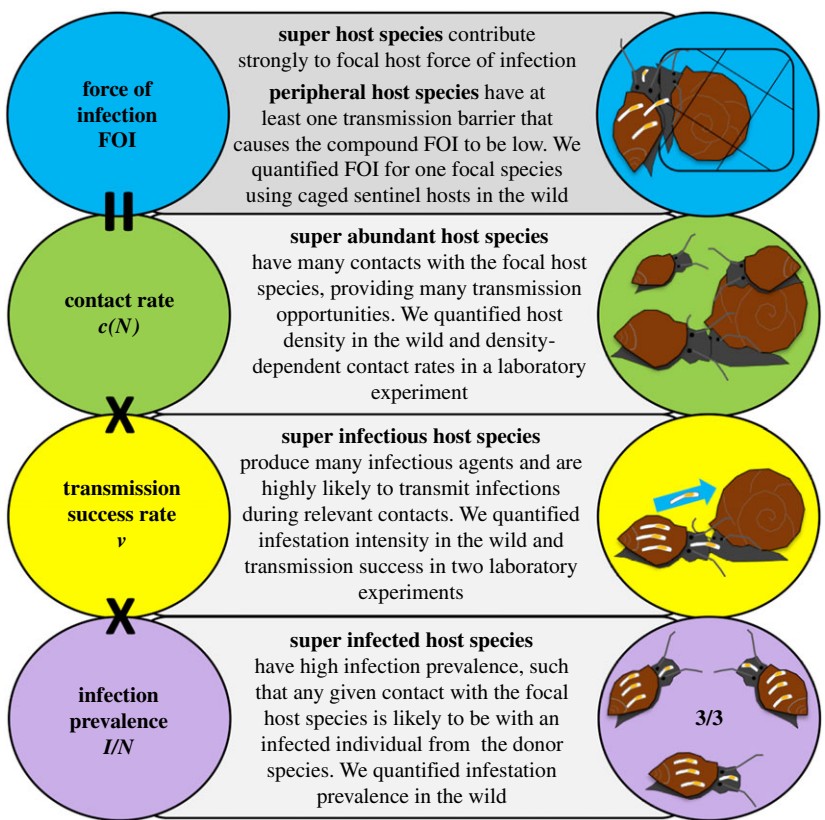

**Figure 1.** The force of infection for any focal host species (FOI_focal) is the rate that individual susceptible hosts ($S_{focal}$) become infected. The FOI is given by the sum of the contributions to transmission by each host species ($i$) in the multi-host community: $FOI_{focal} = \Sigma c(N_i) \times v_i \times (I_i/N_i) \times S_{focal}$, where $c(N)$ is the density-dependent probability that two hosts contact, $I/N$ is the infection prevalence and probability that a given contact is with an infectious individual, and $v$ is the probability of successful transmission given a contact between an infectious and a susceptible individual. These three rate parameters in the FOI equation correspond to three amplifying characteristics that cause host species to contribute strongly to the FOI. By contrast, a single transmission barrier could cause a potentially important host species to be a peripheral host species. In the diagrams, worms indicate that snail hosts are infected, and worm size is exaggerated. (Online version in colour.)

that any given contact with that species is likely to be with an infected individual [9]. These characteristics correspond to the contact rate, transmission success rate, and infection prevalence, which are multiplied together within epidemiological models as the 'force of infection' or net transmission rate [8–10,15,16] (figure 1). Though net transmission rates alone can distinguish key host species from peripheral host species, breaking net rates into their component characteristics can increase mechanistic understanding of disease dynamics. This, in turn, can improve efforts to identify which management methods (e.g. culling, vaccination and test-and-treat) would increase or decrease community transmission for a given key host species [8,9].

Decomposing net transmission rates also reveals that a host species with an amplifying characteristic will not necessarily be a key host species, because epidemiological processes compound. For example, despite having high contact rates and high infection prevalence, a host species might have low net interspecific transmission rates if it has very low infectiousness. In this way, a potentially important host species could be neutralized by one strong transmission barrier. For example, though bank voles and abundant wood mice share the same general habitat and often the same burrows, cowpox virus is rarely transmitted between the two species [17]. However, few studies have identified what causes interspecific transmission barriers because the necessary manipulative experiments are usually prohibitively difficult [8–10,15].

Furthermore, though interspecific transmission barriers are usually framed as host species characteristics (e.g. host resistance and host avoidance behaviour), symbiont characteristics, such as physiology, behaviour, or ecology, could also create transmission barriers [18]. For example, even when symbionts can use several host species, they may strongly prefer one host species, such that interspecific transmission success rates are low even when other host species are abundant and probably frequently encountered. This is the case for West Nile Virus in birds in North America, where mosquitos preferentially feed on particular bird species. Relatively rare, but preferred, host species are more important for community transmission of West Nile Virus than would be expected, and relatively abundant, but avoided, host species are less important for community transmission than would be expected [5,19]. In this example and most others, the impacts of host preferences on community transmission were studied with regards to disease vectors (e.g. mosquitos and ticks), rather than for parasites or beneficial symbionts themselves. However, strong host preferences exist for many symbionts [18,20–22], and these preferences could cause potential key host species to contribute more or less than expected to community transmission. Here, we used an experimentally tractable multi-host system to decompose interspecific transmission into its component parts, revealing host preferences as a strong symbiont-driven transmission barrier that would have otherwise remained unknown.

We focused this study on *Chaetogaster limnaei*, an ectosymbiotic oligochaete that infests at least 16 aquatic snail genera in 10 families [23]. Snails can be harmed by or benefit from their *C. limneai* worms [24–26], depending on the context (i.e. the presence of trematode parasites of the snails), and thus we refer to these worms as symbionts. Symbiont transmission occurs predominantly through direct contact among snails [27], and as worms spread host-to-host through snail host communities in seasonal epidemics [24,28], symbiont prevalence and infestation intensity can vary substantially among snail host species [28–30]. We expected that snail species with high densities, high infestation prevalence, and/or high infestation intensities could be key host species with high interspecific transmission rates (figure 1). However, when we quantified net interspecific transmission rates between two such snail species in the wild, we found that net interspecific transmission rates were low. Therefore, we decomposed the net interspecific transmission rate using manipulative contact rate and transmission success rate experiments in the laboratory. We found that low interspecific transmission success created a strong transmission barrier for this generalist symbiont, and we discuss how to find similar interspecific transmission barriers in other systems.

## 2. Methods

### (a) Study system and design

*Chaetogaster limnaei* has a near global distribution [23,28–30]. In temperate regions, the worms asexually reproduce on snail hosts from spring to autumn. Although worms can survive for short periods when removed from snail hosts, they have high mortality off the host, even when traversing short distances in laboratory environments without predators [27]. Correspondingly, in laboratory experiments, they will not leave living hosts to disperse to other nearby hosts unless hosts are in direct contact [27]. In the late autumn or winter, *C. limnaei* may sexually reproduce and form cocoons that overwinter, but little is known about this life stage, including whether cocoons remain attached to estivating snails or overwinter on substrate [28,31]. In our region in western Virginia, USA, neither sexually reproductive individuals nor cocoons have been observed, and thus we studied asexual individuals transmitted via direct host contacts.

Though *C. limnaei* has a broad host distribution, two snail genera (*Helisoma* and *Physa*) have been the focus for most prior experiments and are especially abundant in ponds across North America. Therefore, for all field and laboratory studies, we used *Helisoma trivolvis* snails as the focal host species (i.e. the species in which we quantified infestation rates) and *Physa gyrina* as the alternative host species. We especially focused on interspecific transmission from *P. gyrina* to *H. trivolvis* (alternative to focal host transmission; hereafter alternative–focal), which complements prior work that quantified intraspecific transmission among *H. trivolvis* snails (focal to focal host transmission; hereafter focal–focal) [27,32]. We used a sentinel approach to quantify the rate that uninfested focal hosts (*H. trivolvis*) that had been raised in the laboratory became infested when caged in the field. We then decomposed the estimated interspecific transmission rate using two laboratory experiments that quantified interspecific contact rates and interspecific transmission success.

### (b) Interspecific symbiont transmission in the field

We determined how infestation risk varied for our focal host species (*F*) between April and September 2013 in a single pond

in Montgomery County, Virginia. We quantified focal host infestation risk (the proportion of sentinel hosts that became infested) by placing laboratory-reared, uninfested focal hosts in field enclosures, where they could directly contact focal and alternative snail species living in the pond through the mesh enclosures. We conducted eight enclosure trials, each lasting approximately one week, at biweekly or monthly intervals (see exact dates in the electronic supplementary material), with enclosures located at the same randomly selected 1 m$^2$ sites in the pond for each trial. Snail infestation status was determined via laboratory dissections within 24 h after transporting snails from the field enclosures to the laboratory in individual 50 ml centrifuge tubes.

Concurrent with the enclosure trials, we also quantified natural variation in the density and infestation prevalence of two wild host species (*P. gyrina* and *H. trivolvis*) at the same eight sites. We collected snails by haphazardly placing up to five quadrats around each 1 m$^2$ site and collecting all snails within the boundary (see the electronic supplementary material). Snail infestation status and infestation intensity were determined via laboratory dissections within 24 h after transporting snails from to the laboratory in individual 50 ml centrifuge tubes. Some of these survey data were previously published in a separate study describing how intraspecific (focal–focal) transmission increases nonlinearly with focal *H. trivolvis* density owing to non-instantaneous contact handling times [32]. Here, we include the data from wild, alternative hosts (*P. gyrina*), to our knowledge for the first time.

A multi-host transmission function can be used to distinguish between the contributions of intraspecific and interspecific transmission to focal host force of infection (FOI$_F$). Note that following mathematical conventions, the FA subscript designates transmission from alternative hosts to focal hosts throughout. We assumed that FOI$_F = \beta_{FF}N_F^{kFF}(I_F/N_F) + \beta_{FA} N_A^{kFA}(I_A/N_A)$, where $\beta_{FF}$ and $\beta_{FA}$ (estimated from the statistical model) are the intra- and interspecific transmission rates, $N_F$ and $N_A$ are the focal host and alternative host densities, $I_F$ and $I_A$ are the infested focal host and infested alternative host densities, and $k_{FF}$ and $k_{FA}$ (estimated from the statistical model) are flexible, unitless density-dependent parameters that allow intraspecific and interspecific transmission to be linear increasing functions of host density ($k = 1$, density-dependent transmission), nonlinear functions of host density ($0 < k < 1$) or independent of host density ($k = 0$, frequency-dependent transmission) [13,32]. Comparing $\beta_{FA}$ to $\beta_{FF}$ allowed us to compare the relative roles of inter- and intraspecific transmission to net focal host infestation risk. However, when fitting a multi-host epidemiological model to this short time series (an eight-point time series summarizing data from 1600 wild and laboratory-reared snails), the number of parameters approached the number of data points. Therefore, while fitting this model can provide insights, exact quantitative estimates should be evaluated with some caution.

Given the multi-host transmission function, the predicted proportion of focal hosts infested by the end of an enclosure trial is given by $1 - \exp(-\text{FOI}_F \times t)$, where $t$ is the duration of the enclosure trial in days [32]. We fit this model to our paired field enclosure and cross-sectional survey datasets using a Bayesian framework with non-informative uniform priors, where we assumed that transmission rates ($\beta_{FF}$ and $\beta_{FA}$) were between 0 and 2 (with 2 being an extremely high value) and that density-dependence parameters ($k_{FF}$ and $k_{FA}$) were between 0 and 1 (see above). The model fitting was conducted with package 'R2jags' [33], using three Markov chain Monte Carlo (MCMC) chains run for 30 000 iterations each, a burn-in of 15 000 iterations, and random starting values for all parameters. Model fits were assessed by examining plots of Pearson's residuals, model predictions, and convergence, and by confirming that all Gelman-Rubin statistics were less than 1.01.

## (c) Interspecific contact rate experiments

We next performed a laboratory contact rate experiment to determine whether low interspecific contact rates could explain the low interspecific transmission rates estimated from the field survey. We previously performed a similar experiment to quantify the relationship between intraspecific *H. trivolvis* contact rates and density [32]. For this study, we quantified how the number of interspecific contacts that individual, focal *H. trivolvis* snails had with alternative host *P. gyrina* snails varied with *P. gyrina* density. We varied alternative host densities along a realistic gradient based on our field observations: 1, 2, 3, 4, 5, 6, 7, 8, 9, 10, 11, 12, 13, 14, 16 or 18 alternative hosts, in 420 cm$^2$ (28 × 15 cm) plastic containers containing 2 l of well water. There were three replicates of each of the 16 alternative host density treatments (48 tanks), and we performed these replicates over six trial days of eight containers each, with even representation of treatment groups in each trial. A few alternative hosts (3 out of 417) were lethargic or dead on the morning of their trials, and we excluded those individuals and adjusted the density treatments to reflect their absence. We added three focal host snails to each container.

All snails were collected in 2015 from the same 1 m$^2$ sites used to quantify focal host FOI in the field. We removed any snails that were shedding trematode cercariae on the day that they were brought into the laboratory, because trematodes may affect snail behaviour [34–36]. The remaining snails were randomly assigned to treatment groups and trials. All snails were painted the day before a trial with a unique two dot colour code using Sally Hansen Insta-dry nail polish (*sensu* [27]).

Just before each trial began, we added loose periphyton to the bottom of all experimental arenas to encourage normal snail foraging and movement behaviour. Periphyton was originally collected from sticks and rocks in a natural pond, which were gently agitated in a bucket of pond water. After filtering out larger particles, the periphyton was aliquoted into larger plastic containers containing dechloraminated water, a surplus of nutrients ($NH_4NO_3$ and $KH_2PO_4$) and one layer of submerged 5.08 x 5.08 cm ceramic tiles. The tiles were left under full-spectrum fluorescent lights for several weeks to promote algal growth, until the tiles were entirely covered with periphyton. Before each trial, the periphyton from two ceramic tiles was added to the experimental container by gently shaking the tiles in the water and allowing the periphyton to settle randomly to the bottom of the container.

After adding periphyton, snails were then added to their appropriate containers 1.5 h before the observation period to acclimate, and then each observation period lasted for 45 min. Each time there was a contact between snails, we recorded the start time of the contact, the individuals involved in the contact, and the end time of the contact. If three or more snails joined together into one cluster, we recorded all unique pair combinations as contacts. All observation periods were also video recorded so that we could confirm details, as needed. Snails in the same container were not independent; for instance, one particularly active alternative *P. gyrina* could cause all three focal *H. trivolvis* in one container to have high interspecific contact rates. Therefore, we used the mean number of interspecific contacts per focal snail per container as the response variable for the contact rate function.

We previously found that a Holling Type II functional response provided a good description of the asymptotic relationship between intraspecific focal *H. trivolvis* contact rates and focal host density [32], where the 'handling time' was defined as the duration of the contact interaction: number of contacts = (encounter rate × total time × density)/(1 + encounter rate × contact handling time × (density − 1)). We fit the same Holling Type II functional response to the interspecific contact rate data, and then we compared the best-fitting interspecific model to our previously published intraspecific model to see if their overall functional shapes, estimated encounter rates ($e_{FF}$ and $e_{FA}$) or estimated handling times ($H_{FF}$ and $H_{FA}$) were different. We fit these models in a Bayesian framework using three MCMC chains with 30 000 iterations each and a 15 000 iteration burn in. We assumed that the mean interspecific contact rate per container was normally distributed, and we used uninformative uniform priors for the interspecific encounter rate (0–1) and interspecific contact handling time (0–44 min). The model fitting was conducted with package 'R2jags' [33], and model fits were assessed as described above.

## (d) Interspecific transmission success experiments

Finally, we quantified the rates at which symbionts dispersed from infested donor hosts to uninfested receiver hosts of the same species (focal *H. trivolvis* to focal *H. trivolvis*) versus the rates at which symbionts dispersed from donor hosts to receiver hosts of a different species (alternative *P. gyrina* to focal *H. trivolvis*). In both treatments, symbionts were added to previously uninfested donor snails (focal or alternative hosts) that were raised in the laboratory. This donor focal snail was then placed in a container small enough to ensure contacts with an uninfested receiver focal host that was also raised in the laboratory. The response variable was the proportion of symbionts that dispersed from the donor to the receiver snail during the experiment.

The symbionts added to the donor hosts were collected from wild source snails from the same pond used in the field FOI trials, and we did not know whether the wild source host species (focal *H. trivolvis* or alternative *P. gyrina*) would affect transmission rates. Therefore, we performed two separate experiments: one with symbionts sourced from focal *H. trivolvis* (focal-sourced) and one with symbionts sourced from alternative *P. gyrina* (alternative-sourced) (electronic supplementary material, figure S1). Because we were especially interested in quantifying interspecific transmission rates for the first time in this system, we performed more replicates of the interspecific transmission treatments (i.e. alternative donor to focal receiver). In particular, we performed 20 replicates of the focal-sourced intraspecific treatment (focal–focal$_{F-sourced}$), 21 replicates of the focal-sourced interspecific treatment (alternative–focal$_{F-sourced}$), 40 replicates of the alternative-sourced intraspecific treatment (focal–focal$_{A-sourced}$) and 49 replicates of the alternative-sourced interspecific treatment (alternative–focal$_{A-sourced}$). As *C. limnaei* can rapidly asexually reproduce, some experimental units ended up with more total symbionts than were initially added. Therefore, instead of using final symbiont counts on receiver snails as the response variable, we evaluated whether the proportion of symbionts that dispersed from the donor to receiver snail was affected by the treatment group (intraspecific versus interspecific pairing), the symbiont source (focal-source versus alternative-source of symbionts; electronic supplementary material, figure S1), and/or the interaction between the treatment group and source using a binomial generalized linear model with a logit link.

For both experiments, we separated laboratory-raised, uninfested snails into individual 150 ml plastic cups containing approximately 50 ml of well water 2 days before the experiment. Focal hosts (always *H. trivolvis*) were randomly assigned to treatment groups and replicates, and then focal hosts in the intraspecific treatment (focal–focal) were randomly assigned to be donor versus receiver snails. Alternative hosts were randomly assigned to replicates in the interspecific (alternative–focal) treatment group. The day before the experiment, the receiver snail in each replicate was painted with a single dot of nail polish (*sensu* [37]), and 10 *C. limnaei* symbionts collected by pipette from the appropriate wild source snail species were added to each of the cups containing donor hosts (see the electronic supplementary material) [27]. All experimental donor and receiver snails were fed Spirulina fish food and left overnight to allow time for the symbionts to find and attach to the donor hosts.

On the morning of each experiment, the donor and receiver snails from each replicate were combined in a single 150 ml plastic cup (hereafter 'experimental cup'). Snails were initially placed such that they were not touching. In the first experiment (alternative-sourced), all replicates were observed for 1 h after snails were placed together in experimental cups. During that time, all donor–receiver pairs had contacted at least once, confirming that the experimental cups were small enough to ensure contacts. Because all snail pairs had contacted, and because in a previous experiment, we found that approximately 16% of *C. limnaei* disperse from donor focal *H. trivolvis* to receiver focal *H. trivolvis* in 1 h [27], we assumed that 1 h was enough time for potential transmission. We therefore separated the donor and receiver snails into their individual containers after 1 h and began dissections to quantify the number of symbionts per snail. However, after dissecting all snails from eight replicates in each of the intraspecific (focal–focal) and interspecific (alternative–focal) treatment groups, we were surprised to find that only one symbiont was recovered from a receiver snail. We therefore placed the remaining donor and receiver snails together in their experimental cups for an additional 17 h. We discuss the initial eight replicates here, but we only included the 73, 18 h replicates in our statistical models. When we ran the second experiment (focal-sourced), we paired all donor and receiver snails in their experimental cups for 18 h prior to separating and dissecting all snails.

While dissecting snails in the alternative-source experiment after the full 18 h trial, we noticed that we were not recovering as many symbionts as we had added to snails the day before, which was unusual. To understand where these symbionts had gone, we checked the individual donor cups, receiver cups, and experimental cups and recorded any symbionts that were not attached to snails. These symbionts could have (i) failed to add to the donor snail prior to the experiment (donor cups); (ii) dispersed from the donor to the receiver snail and then fell off or left after the receiver snail was placed back in its individual cup (receiver cup); or (iii) fallen off either snail during the experiment (experimental cup). After summing the total symbionts recovered on snails or in cups, we used Poisson generalized linear models (GLMs) with log links to determine whether the total number of surviving symbionts varied across treatment groups, using the R package 'MASS' [38]. Model fits were assessed visually by examining plots of model predictions and Pearson's residuals.

# 3. Results

## (a) Interspecific symbiont transmission in the field

In the field, alternative *P. gyrina* densities ranged from 0 to 24 snails 0.1 m$^{-2}$ and peaked earlier than focal *H. trivolvis* densities (figure 2b). For both host species, the prevalence of *C. limnaei* infestation peaked late in the season above 70% and was followed by a subsequent decline (figure 2d). Symbiont infestation intensity (worms per snail) tended to be higher in focal *H. trivolvis* than alternative *P. gyrina* during most weeks later in the summer (figure 2c), when alternative host densities were low. When alternative host densities were high, the rates that caged focal hosts became infested (FOI$_{focal}$) were very low, and when focal host densities were high, forces of infection were high (figure 2).

Correspondingly, adding interspecific transmission to the force of infection model did not improve explanatory power, where the multi-host model that included interspecific (alternative–focal) transmission had a somewhat higher deviation information criteria (DIC) value (pD = 2.46, DIC =

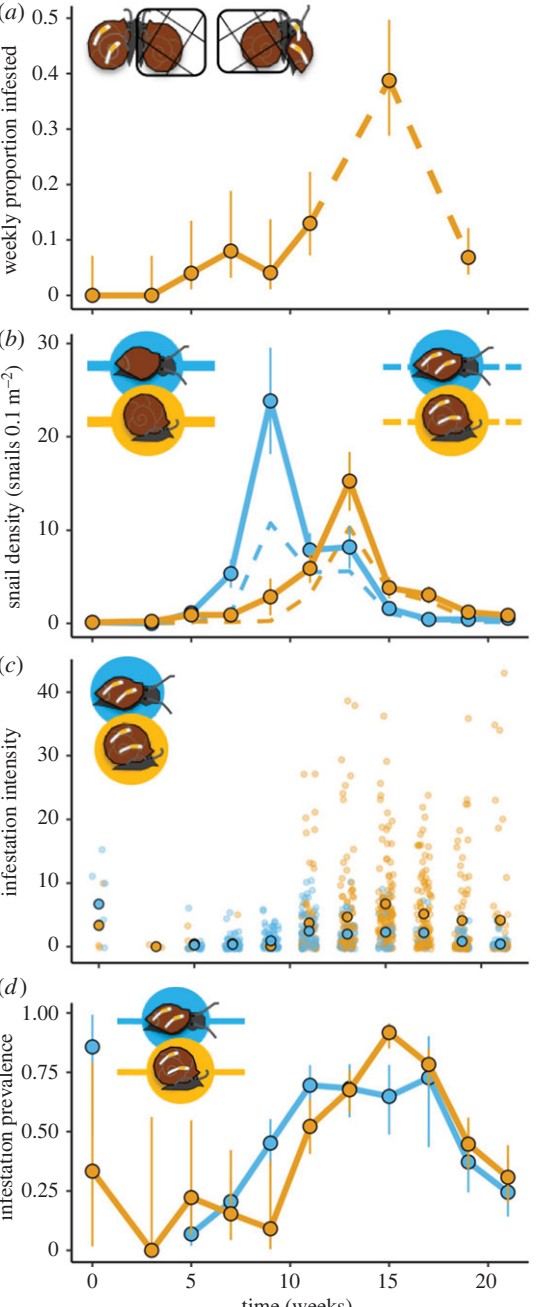

**Figure 2.** (a) The weekly infestation rate, defined as the proportion of laboratory-reared, uninfested sentinel hosts from the focal host species (*H. trivolvis*) that became infested by symbionts during each of eight field enclosure trials. The vertical bars delineate 95% asymmetrical binomial confidence intervals for each FOI. Note that there were no enclosure trials on weeks 13 or 17. (b) The wild alternative *P. gyrina* (blue) and focal *H. trivolvis* (orange) snail densities in the pond each week, where the vertical bars are 95% Poisson confidence intervals for each observation. The dashed lines show infested snail densities. (c) The number of worms per wild snail in the pond each week, with the mean number of worms per snail overlaid in a black-outlined circle. (d) The symbiont infestation prevalence in wild alternative *P. gyrina* and focal *H. trivolvis* snails, shown as a proportion with vertical bars delineating 95% asymmetrical binomial confidence intervals. The time scale starts on 20 April 2013 (week 0) and ends on 15 September 2013 (week 21). (Online version in colour.)

39.48) than the previously published single-host models that only included intraspecific (focal–focal) transmission (pD = 1.98, DIC = 38.06) [32]. Furthermore, the model estimated a very small role for interspecific transmission. In particular, though the 95% credible interval for the

Proc. R. Soc. B **289**: 20220084

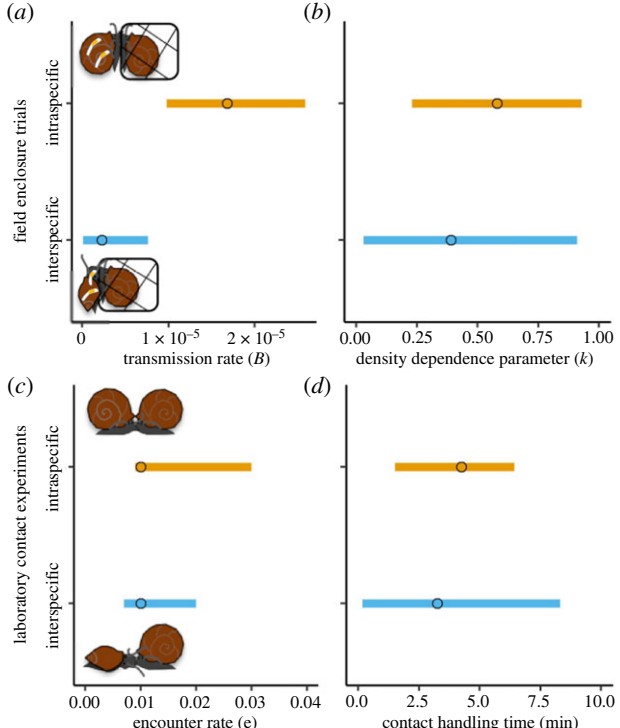

**Figure 3.** In the field enclosure trials (*a,b*), estimated interspecific (alternative–focal) transmission rates from field hosts to caged sentinel hosts were an order of magnitude lower than estimated intraspecific (focal–focal) transmission rates, and 95% credible intervals did not overlap. By contrast, in the laboratory contact rate experiment (*c,d*), estimated interspecific (alternative–focal) encounter rates and contact handling times were nearly identical to intraspecific (focal–focal) encounter rates and contact handling times from a prior experiment [32], and 95% credible intervals overlapped, suggesting that interspecific contact rates are as common as intraspecific contact rates for any given host density. (Online version in colour.)

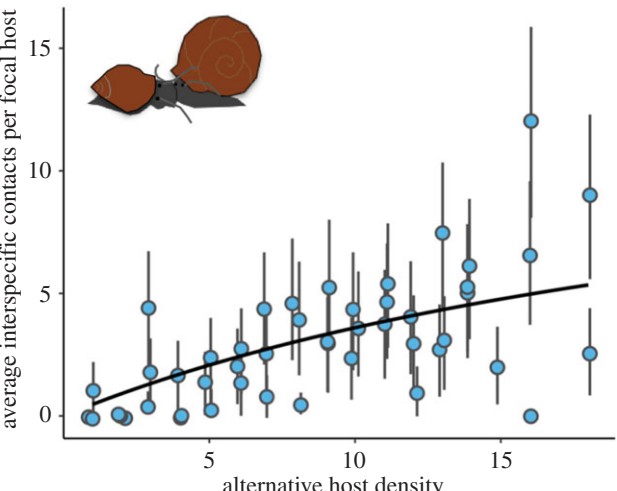

**Figure 4.** The mean number of interspecific contacts (alternative–focal) made by the three focal host snails (*H. trivolvis*) in each container during 45 min observation periods in the laboratory contact rate experiment, where the vertical bars delineate 95% Poisson confidence intervals. Points are jittered slightly on the *x*-axis to aid visualization. The black line is the best-fitting interspecific contact rate function. (Online version in colour.)

interspecific transmission rate in the multi-host model did not overlap zero (figure 3), the interspecific transmission rate was at least one order of magnitude smaller than the intraspecific rate. Note that because the interspecific transmission rate was so low, the credible interval for the estimated unitless density-dependence parameter ($k_{FA}$) included almost all values between 0 and 1 (figure 3).

## (b) Interspecific contact rate experiment in the laboratory

We observed 559 interspecific contacts between focal *H. trivolvis* and alternative *P. gyrina* snails. Individual focal hosts had anywhere from 0 to 16 interspecific contacts during a 45 min observation period, and interspecific contact rates increased nonlinearly with alternative host density (figure 4). When comparing 95% credible intervals, the interspecific (alternative–focal) encounter rate (0.012 encounters min$^{-1}$) estimated by the Holling Type II functional response was no different from the intraspecific (focal–focal) encounter rate (0.014 encounters min$^{-1}$) that was previously published (figure 3) [32]. Similarly, the contact handling time for interspecific (alternative–focal) contacts was estimated to be 3.25 min in the interspecific Holling Type II contact rate function, which was not statistically different from the intraspecific (focal–focal) contact handling time previously published (figure 3) [32]. Overall, the contact rate function describing

how interspecific (alternative–focal) contact rates varied with alternative host density was indistinguishable from the contact rate function describing how intraspecific (focal–focal) contact rates varied with focal host density.

## (c) Interspecific transmission success experiment in the laboratory

The rates of intraspecific and interspecific transmission success that we observed in the laboratory depended somewhat on interspecific versus intraspecific host treatments, but more so on whether the symbionts used in the experiment were sourced from alternative *P. gyrina* or focal *H. trivolvis* snails. In the focal-source trial, interspecific (alternative–focal) transmission rates were actually higher than intraspecific (focal–focal) transmission rates (83% versus 32%), as measured by the proportion of symbionts on the receiver host at the end of the trial (binomial GLM, d.f. = 107; main interspecific treatment effect ± s.e. = 2.17 ± 0.25, $p < 0.001$; figure 5*a,b*). This result contradicted our prediction and was instead consistent with symbionts preferentially dispersing towards receiver host species that matched their original wild host species (i.e. leaving novel alternative donor host *P. gyrina* for more familiar focal *H. trivolvis*). In comparison to the focal-source experiment, transmission rates were very low in the alternative-source experiment (main alternative-source effect ± s.e. = −1.66 ± 0.38, $p < 0.001$); just 3% and 8% of the symbionts dispersed to the receiver host during the 18 h trial (figure 5*c,d*). Furthermore, the treatment effect reversed, such that interspecific (alternative–focal) transmission rates were lower than intraspecific (focal–focal) transmission rates (treatment × source interaction term ± s.e. = −3.10 ± 0.53, $p < 0.001$). Across the two experiments, transmission success was high in the treatment that best represented intraspecific transmission in the field (32%; focal–focal$_{F-source}$; figure 5*a*)—approximately 10 times higher than in the treatment that best represented interspecific transmission in the field (3%; alternative–focal$_{A-source}$; figure 5*d*).

In contrast with our previous experimental work in this system, which focused only on intraspecific transmission

among focal *Helisoma* snails, we recovered many *C. limnaei* from the bottoms of cups that were not attached to snails during the alternative-source experiment. Of the total recovered symbionts, less than 1% (3 out of 396 worms) were unattached in the cups in the interspecific treatment where the donor snails were alternative *P. gyrina* (same as alternative-source snails), whereas 24% (38 out of 158 worms) were unattached in the intraspecific treatment cups where the donor snails were focal *H. trivolvis* (donor species differed from alternative-source species). In the second case, most symbionts were recovered from the donor cup (87%; 33 out of 38 worms), as opposed to the experimental cup (8%; 3 out of 38 worms) or the receiver cup (5%; 2 out of 38 worms), suggesting that the symbionts never infested *H. trivolvis* donor snails during experimental additions. Furthermore, when summing the total symbionts in cups or on snails, we found that 40.5% of the originally added 730 symbionts were completely missing at the end of the experiment. These worms either died and degraded beyond recognition within the 18 h period, or they were ingested by snails after they died. These mortality events were twice as likely in the intraspecific (focal–focal) treatment as in the interspecific (alternative–focal) treatment (four versus eight remaining *C. limnaei*, on average; Poisson GLM, $p < 0.001$, d.f. = 87). Therefore, we conclude that a large proportion of the symbionts never attached to the donor snail if it was a different species from the wild source hosts, and among those symbionts that never attached, many died. This is an unexpected and strong demonstration of host preference, where symbionts with available hosts died before adding to an unfamiliar host species.

## 4. Discussion

By breaking the transmission process down into its component parts, we found that a host species with key host characteristics (super abundant and super infected; figure 1) was not a major source of interspecific transmission owing to a strong and previously unrecognized transmission barrier. In the wild, the two host species had similar density, prevalence of infection, and symbiont infestation intensity. However, estimated interspecific (alternative–focal) transmission rates were an order of magnitude lower than intraspecific (focal–focal) transmission rates to sentinel hosts. Low interspecific transmission rates could not be explained by behavioural or spatial segregation that prevented the two host species from contacting; the two species were commonly observed contacting in the wild, and interspecific contact rates were indistinguishable from intraspecific contact rates in controlled laboratory experiments. Instead, transmission success during any given interspecific contact was low because symbionts showed a strong preference for their current host species. After 18 h in a small cup that forced repeated contacts between hosts, only 6% of symbionts dispersed between host species, in contrast with 32% within species. This transmission barrier reduced interspecific transmission rates to almost undetectable levels in the wild, despite frequent interspecific transmission opportunities.

Why would a host generalist like *C. limnaei* have strong host preferences? *Chaetogaster limnaei* cannot swim and become tangled in debris when not attached to a host snail, and thus many oligochaetes die if their current host dies, even when another suitable host is nearby [27]. In fact, in

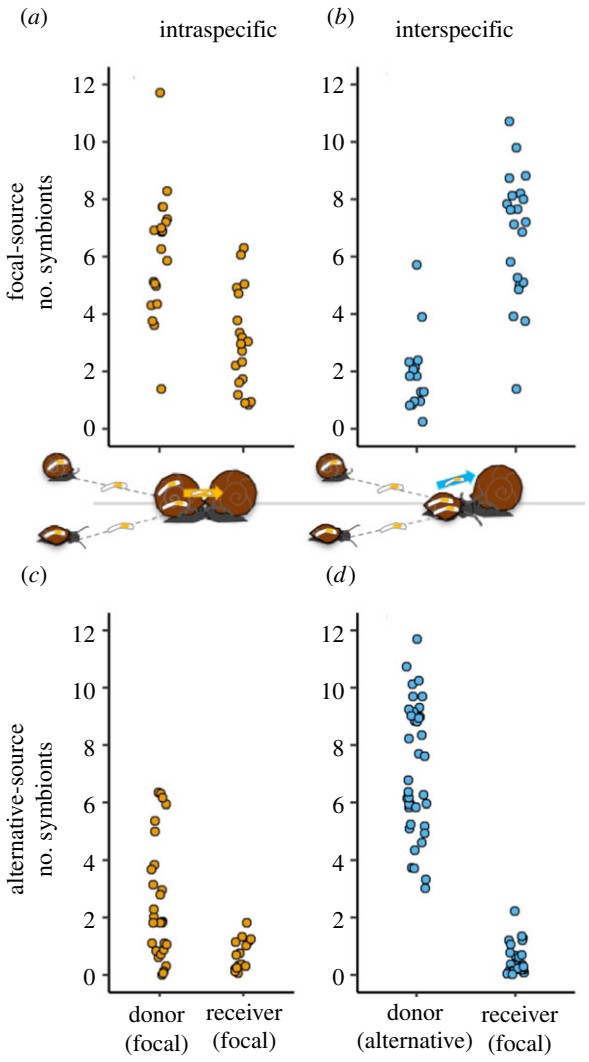

**Figure 5.** Symbiont transmission success rates, where individual points show the number of symbionts remaining on the donor snail versus the number that dispersed to the receiver snail by the end of the experiment. (*a*) Best represents the wild intraspecific transmission scenario (32% transmission), where symbionts were sourced from the same species as the donor and receiver hosts (focal-source with focal–focal transmission). (*d*) Best represents a natural interspecific transmission scenario (3% transmission), where symbionts were sourced from the alternative donor species (alternative-source with alternative–focal transmission), which differed from the receiver species. Points are jittered slightly on the *x*-axis to aid visualization. Like the panel labels, the diagrams in the middle of the figure show whether the column refers to intraspecific or interspecific transmission and whether the row refers to focal or alternative sources of symbionts (also see diagram in the electronic supplementary material, figure S1). (Online version in colour.)

this study, many oligochaetes died in the absence of natural predators, and this mortality was only observed in treatments where the wild source species was different from the donor species (i.e. host availability did not match preferences). Because being free-living is perilous for the oligochaetes, *C. limnaei* probably experiences opposing selection pressures. On one hand, it may be beneficial for the oligochaetes to strongly orient towards chemical cues from their current host or the most abundant host species in the habitat [30,39], so that they can rapidly re-find the host if dislodged. The results of our transmission success experiment could be consistent with frequency-dependent preferences, where oligochaetes preferred source hosts that were most abundant in the pond at the time of each experiment. On the other

hand, individuals (asexual clones) that manage to disperse between species might also be highly successful, especially where the relative availability of different snail species changes throughout the season (figure 2, [24,28,29]). Correspondingly, despite strong preferences, *C. limnaei* taken from one host species can habituate to a new host species in the laboratory [30,39]. Like *C. limnaei*, many vectors and ectoparasitic species might experience a similar tension between taking advantage of any available host and specializing on particular host species cues to increase host finding success. This trade-off may be limiting disease spread in many systems, because when host preferences are so strong that symbionts choose not to disperse to competent hosts, total transmission could be reduced relative to the maximum possible transmission.

Strong host preferences and specificity occur in many systems [8,40–42], but as we show here, they can be difficult to identify. We discovered low net interspecific transmission rates in the field, but we could not use that correlational study to identify the causative barrier to transmission because the components of the transmission rate are multiplied together and thus unidentifiable (figure 1). Instead, we identified strong host preferences as the barrier using a bottom-up experimental approach that decomposed net interspecific transmission rates. More work like this is needed [10], but it will not be feasible in many systems where host species are difficult to observe, manipulate, or experimentally infect. Another possible approach is to look for symbiont population genetic structuring (preferences) or species differentiation (specificity) among host species [41,43]. For example, *Bartonella* species infect multiple rodent host species in the UK, but there tend to be unique genetic variants circulating in each rodent species, suggesting interspecific transmission is rare [41]. For *C. limnaei*, extensive morphological analyses and limited existing genetic comparisons suggest that there is only a single *C. limnaei* species in North America [23,30]. This is not surprising, given that our best-fitting interspecific transmission model estimated that interspecific transmission rates were very low, but still present. However, there could be notable genetic structuring among *C. limnaei* from different species, and this is a promising avenue for future research. Genetic analyses alone cannot necessarily disentangle whether interspecific transmission rates are low owing to limited interspecific contacts or symbiont preferences.

However, when contact rates are known or expected to be high, genetic analyses might prove more feasible for identifying host preferences or specificity in the many systems where experimental manipulation is not possible.

Understanding how symbionts spread within and between species is important, because these processes can be used to augment spread of beneficial symbionts or control transmission of parasites that protect people, domestic species, or wildlife [1,2,6,11]. For instance, abundant species will often act as disease reservoirs, and thus abundant species will often be disease control targets [3,44]. Though identifying key host characteristics like these can be helpful [7–9], it might be equally important to think about existing transmission barriers. These transmission barriers are not necessarily permanent, and a system perturbation could unleash superspreading potential that was previously unidentified by quantifying net transmission rates alone. Therefore, as we continue to quantify the relative contributions of each host species to symbiont maintenance in host communities, it is important to remember that host and symbiont characteristics compound to increase or decrease a host species' overall contribution to interspecific transmission. Determining both the magnitude of interspecific transmission and why the magnitude is relatively large or small will better enable us to predict and control transmission.

**Data accessibility.** The datasets, R scripts and metadata supporting this article are available from the Dryad Digital Repository: https://doi. org/10.5061/dryad.hmgqnk9jw [45].

**Authors' contributions.** S.R.H.: conceptualization, data curation, formal analysis, funding acquisition, investigation, methodology, software, supervision, visualization, writing—original draft, writing—review and editing; C.M.M.: data curation, investigation, methodology, writing—review and editing; L.K.B.: conceptualization, funding acquisition, writing—review and editing; J.M.W.: conceptualization, formal analysis, funding acquisition, methodology, writing—review and editing.

All authors gave final approval for publication and agreed to be held accountable for the work performed therein.

**Competing interests.** The authors declare no competing interests.

**Funding.** This work was supported by the National Science Foundation (grant nos. DEB-1556729, DEB-1501466, DEB-0918960, DEB-0918656) and Virginia Tech's Graduate Student Assembly's Graduate Research Development Program.

**Acknowledgements.** We thank J. Walters and B. Brown for their helpful comments on initial drafts.

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
