## [Peer Review File · Proceedings of the Royal Society B: Biological Sciences]

Review History

RSPB-2021-1224.R0 (Original submission)

Review form: Reviewer 1

Recommendation

Major revision is needed (please make suggestions in comments)

Scientific importance: Is the manuscript an original and important contribution to its field?

Good

General interest: Is the paper of sufficient general interest?

Acceptable

Quality of the paper: Is the overall quality of the paper suitable?

Good

Is the length of the paper justified?

Yes

Should the paper be seen by a specialist statistical reviewer?

No

Do you have any concerns about statistical analyses in this paper? If so, please specify them explicitly in your report.

Yes

It is a condition of publication that authors make their supporting data, code and materials available - either as supplementary material or hosted in an external repository. Please rate, if applicable, the supporting data on the following criteria.

Is it accessible?

Yes

Is it clear?

Yes

Is it adequate?

Yes

Do you have any ethical concerns with this paper?

No

Comments to the Author

This manuscript presents field and lab experimental results that seek to understand the role of host preference in determining rates of cross-species transmission for multi-host parasites. The ms is well written, with the underlying logic clearly laid out, and the experiments a well thought through, following a logical progression, and are sensibly analysed. Overall the results are clear – despite this symbiont being a host generalist, and able to infest both host species studied, there are clear preferences for the ‘natal’ host, thereby limiting rates of cross-species transmission, even when provided with the opportunity. As such this work provides a neat refinement of our current understanding of multi-host transmission dynamics – such preferences are probably well enough known to be important for vector-borne pathogens, but this work highlights similar host choice effects for parasites with active transmission stages that have the capacity to introduce an element of host choice into the transmission dynamics.

I have a few comments that I would like to see the authors consider:

- Line 87: “Symbiont transmission occurs predominantly through host contact...” – it might be useful to further clarify/quantify this for those of us who are less familiar with the system. The off-host risks to the parasite are picked up later on in the Discussion, but some clearer statement here would help us understand better the importance of true host-to-host contact, as measured in the subsequent experiments.
- Line 118 – how much are the mesh bags likely to affect transmission? If (as above) transmission really is through contact, could the bags be reducing transmission rates in the field? And is this likely/possible to differ for inter- and intraspecific contact rates?
- In the field experiment the snail community (densities and infection prevalences of both host species) were “surveyed”. I think we need more information here – presumably these data were used to give the IF, NF, IA, NA parameters in the subsequent FOI equation – in which case it is important to know how they were estimated and how intensive the surveying was; estimating the beta and k values may well be quite sensitive to these input parameters, so it would be good to have a better handle on how reliable and accurate they are.
- Line 119 – the response variable from the field experiments was the proportion of experimental snails that got infected – that’s a perfectly sensible response variable to look at, but I’m not comfortable with that being called the Force of Infection – “proportional prevalence” (at the group level) or probability of “being infected” (at the individual level) would seem perfectly fine terms. The FOI though is really something a bit different, though related: for an SI-type framework it would be something like $FOI = -\ln(\text{proportion infected})$. I’d suggest either the authors rephrase what they call it on Line 119 or, perhaps preferable to keep the connection with how they subsequently break it down, I would use that transformation in the subsequent model

fitting (ie, use $-\ln(\text{proportion infected})$) as the response being fitted to for the equation on Line 133).

- Line 237 - I wonder if rather than using 2 GLMs it would be better to use a single GLM for all the data, with a possible Treatment x Source interaction term?

- Figure 2 - it would be interesting to see a plot of host density x infestation prevalence (effectively panels B x D) to give an indication of how density of infecteds changes across the season, and how this relates to infections in the sentinels. Again, I would suggest relabelling the y-axis of panel A to read "Weekly proportion infected" or similar (or transform the data, as suggested above, to make it more directly related to FOI).

- Figure 3 - would be useful to indicate more clearly on the figure which panels relate to the field experiment and which to the lab. It would also be useful to switch them round, since the field results are discussed first in the text.

Review form: Reviewer 2

Recommendation

Major revision is needed (please make suggestions in comments)

Scientific importance: Is the manuscript an original and important contribution to its field?

Good

General interest: Is the paper of sufficient general interest?

Good

Quality of the paper: Is the overall quality of the paper suitable?

Good

Is the length of the paper justified?

Yes

Should the paper be seen by a specialist statistical reviewer?

No

Do you have any concerns about statistical analyses in this paper? If so, please specify them explicitly in your report.

No

It is a condition of publication that authors make their supporting data, code and materials available - either as supplementary material or hosted in an external repository. Please rate, if applicable, the supporting data on the following criteria.

Is it accessible?

Yes

Is it clear?

Yes

Is it adequate?

Yes

Do you have any ethical concerns with this paper?

No

Comments to the Author

Hopkins et al. present an interesting study examining the transmission of a symbiont between two different host species. The authors attempt to tease apart the impact of host density, host infectiousness, and host prevalence on symbiont transmission. The methodologies are interesting and yield some interesting results, but I do have a few concerns regarding data comparisons that are made (more below). The manuscript deals with multiple different species and multiple different experiments that can be hard to keep track of at times. This isn't surprising given the multiple facets of the study and the authors do a good job of setting up the work (especially in the figures which are pivotal and greatly aid in understanding the study), but I think some work still could go in to improving clarity. This is especially true for the first experiment, which I believe needs more details from the previous work included in this manuscript. While the results are certainly interesting, I think the introduction over sells the importance of the study and tries to hype the study beyond what may be defensible. Primarily, many of the results can fall under the umbrella of host preference, host specificity, and host competency, all of which are fairly well described phenomena which I feel this paper could spend more time exploring in the introduction. The results are most certainly of value and worth seeing in print, here or elsewhere, with some fine tuning and careful consideration on revisions. I break down my specific points section by section below, highlighting major and minor points within each section.

Title: I would cut 'strong'. How is strong being quantified? Is it relative to something else? I would also replace superspreader. The word appears little in the actual text and tends to be more a trait of an individual within a species than of a species itself in epidemiological contexts. It seems like host competence or infectiousness would be more specific for this context.

Keywords:

I think some more could be added here. Maybe: host-choice, specificity, species interactions, force of infection... I think you can hit on more elements.

Abstract:

Good

Introduction:

The introduction is overall well written and makes many good points. My main issue is that the points discussed in the introduction feel somewhat disjunct from the discussion and focus largely on the aspects of the first experiment and less so on the other two. I feel host preference and specificity warrant more introduction.

Methods:

My major concerns arise largely from the methods, primarily that I feel some controls may be missing that could provide valuable information.

Experiment 1: Much of this data appears to be coming from a previous experiment (which is fine), but we need some more background on the connection between the previous and current work. Were they done in the same year? If not, what issues arise from comparing these data? Were snail and symbiont densities similar in both years? How was sampling done? etc. Also, were lab experiments done with caged snails? If not, comparing any lab and field results may not be valid. What were cages constructed from? How could this influence snail interactions and symbiont transmission? I am sure much of this is discussed in the other paper, but some of these details seem pertinent (even if in a supplement).

I enjoyed the implementation of the equations. They work out quite intuitively. For readers not mathematically inclined I think a really brief mention on the shape of these curves (mostly asymptotic) would be good. Additionally, mentioning how your experiment generates the parameter estimates (which were measured and which were estimated via Bayesian inference) would be good.

Line 150-151: Is there a specific reason these ranges were chosen?

Experiment 2: Does infestation impact contact rates? If high loads of symbionts alter host mobility, how applicable are your results? Or if load is correlated with size how do you separate this collinearity?

Line 164: I am not sure how to interpret the ..., does this mean 1 2 3 5 7 9 or single snail increments until 14? The associated figure appears to show a data point at 15.

Line 171: Why remove trematode infected snails? I know trematodes can have attracting forces on other snails (See papers by Eliuk et al. Canadian Journal of Zoology and by Gray et al. Foci of transmission). That is a good enough reason, but trematodes are common in snails so might warrant a quick mention.

Line 181: How do the symbionts reproduce? Could symbiont eggs come in on periphyton and alter your experiment? Many trematode eggs remain in sediment until ingested also.

Line 197: Is there a reason to take the mean of the beaker instead of using beaker/ trial as a random factor to explain variance in contact while boosting sample size a bit?

Line 209: Since you have this data can't you just look at the distribution instead of assuming its shape?

Experiment 3: The addition of the terms receiver and donor is A LOT to track for the reader. I suggest in the figure axes to include terms like alternative and focal that the reader has adjusted to to help orient the reader. Overall tracking *Helisoma*, *Physa*, alternative, focal, donor, receiver, sourced became a bit overwhelming and required a lot of back and forth which made a lot of the manuscript feel disjointed.

In this experiment, the authors use symbionts sourced from different snails. But I am wondering if the environmental conditions at the time of collection couldn't have influenced your results. If the snail hosts were at different densities in the environment at the time of sampling, could this have influenced or habituated the symbiont to a certain stimulus? The authors mention that in a previous experiment the symbiont readily attached to the snails they were presented with, but in this experiment, many didn't attach to a host. This would seem to suggest that something abnormal may have been happening. If *Helisoma* densities were really high when symbionts were collected, perhaps symbionts were biased to seek out abundant hosts?

Additionally, were symbionts for this experiment randomized? If symbionts from a certain host were used in this experiment couldn't this generate pseudoreplication error associated with relatedness and symbionts originating from the same host?

Experiment 2 and 3: Why were Focal to alternative transmission experiments (and alternative to alternative) not done? These controls would seem to provide valuable data about interspecific and intraspecific transmission that is unaccounted for here.

Host size consistently arises as a factor in many snail-symbiont interactions. How does your experimental design account/ control for host size?

Results:

Line 294: Are there differences in DIC that are standardized for model support?

Figures are nice and very necessary.

Discussion:

Could hosts be showing some sort of frequency dependent host selection based on the environment they were raised in?

Given the limited genetic work on the symbiont, might you be working with multiple cryptic species that have variable host preferences?

Is there a fitness cost of host switching for a symbiont? How do symbionts reproduce, and could this have an effect on preferred paths of transmission?

Figure 1: Give the definitions on the variables in the equation.

Figure 2C: Go ahead and add units to infection intensity for clarity. Also, this data looks more negative binomial than Poisson, or at least Poisson with overdispersion beyond 1. Consider using a different distribution if it is a better fit.

Could the x axis on all figure 2 be made into a calendar date so this data may be more useful for

judging other environmental factors? And use by others.

Figure 4: Provide a pseudo-r squared for you curve?

Figure 5: The snail images in the center are really useful, but blend in when looking at the data. Mention the diagram specifically in the caption or elsewhere because it's super helpful once you figure out what it means.

Some trails appear to have more symbionts than should have been present (10?). Where did the extra symbionts come from?

This is a really interesting system and I will be interested to see more work come out of further studies. Best of luck! And feel free to contact me via the editor if I can be of any assistance.

Decision letter (RSPB-2021-1224.R0)

07-Aug-2021

Dear Dr Hopkins:

I am writing to inform you that your manuscript RSPB-2021-1224 entitled "Strong host preferences inhibit transmission from potential superspreader host species" has, in its current form, been rejected for publication in Proceedings B.

This action has been taken on the advice of referees, who have recommended that substantial revisions are necessary. With this in mind we would be happy to consider a resubmission, provided the comments of the referees are fully addressed. However please note that this is not a provisional acceptance.

Sincerely,

Dr Sasha Dall

Associate Editor

Board Member: 1

Comments to Author:

Both reviewers agree (and I concur) that this is an interesting, important manuscript. The logic is nuanced and yet clearly presented - this is a complicated study, but it is written with such skill that the arguments are quite easy to follow. The MS tackles an extremely timely topic with implications for the transmission of emerging infectious diseases. There is a lot to be excited about in this MS.

However, the reviewers have identified several weaknesses that must be addressed if we are to consider a revised version of the MS. In particular, please pay attention to the following:

1. **METHODOLOGICAL DETAILS** - We'll need more detail on how the field measurements of snail density and infection prevalence were conducted. At the moment, it would be impossible to replicate these protocols with the information given in the Methods, and we're unable to evaluate the rigor of the snail survey without more detail on how it was conducted. Similarly for the experiments - the MS is very reliant on reference to previous studies. We'll need enough detail on all protocols for a reader to replicate the study without referring to previously published work.
2. **FORCE OF INFECTION** - Please see Reviewer 1's comments regarding the "Force of Infection" term. It may be necessary either to choose different terminology or re-calculate this value.
3. **FRAMING** - Consider Reviewer 2's comments about framing. A lot of literature already exists on host preference, host specificity, and host competency, so please spend more time outlining this context in the Introduction.

Reviewer(s)' Comments to Author:

Referee: 1

Comments to the Author(s)

This manuscript presents field and lab experimental results that seek to understand the role of host preference in determining rates of cross-species transmission for multi-host parasites. The ms is well written, with the underlying logic clearly laid out, and the experiments a well thought through, following a logical progression, and are sensibly analysed. Overall the results are clear - despite this symbiont being a host generalist, and able to infest both host species studied, there are clear preferences for the 'natal' host, thereby limiting rates of cross-species transmission, even when provided with the opportunity. As such this work provides a neat refinement of our current understanding of multi-host transmission dynamics - such preferences are probably well enough known to be important for vector-borne pathogens, but this work highlights similar host choice effects for parasites with active transmission stages that have the capacity to introduce an element of host choice into the transmission dynamics.

I have a few comments that I would like to see the authors consider:

- Line 87: "Symbiont transmission occurs predominantly through host contact..." - it might be useful to further clarify/quantify this for those of us who are less familiar with the system. The off-host risks to the parasite are picked up later on in the Discussion, but some clearer statement here would help us understand better the importance of true host-to-host contact, as measured in the subsequent experiments.
- Line 118 - how much are the mesh bags likely to affect transmission? If (as above) transmission really is through contact, could the bags be reducing transmission rates in the field? And is this likely/possible to differ for inter- and intraspecific contact rates?
- In the field experiment the snail community (densities and infection prevalences of both host species) were "surveyed". I think we need more information here - presumably these data were used to give the IF, NF, IA, NA parameters in the subsequent FOI equation - in which case it is important to know how they were estimated and how intensive the surveying was; estimating

the beta and k values may well be quite sensitive to these input parameters, so it would be good to have a better handle on how reliable and accurate they are.

- Line 119 – the response variable from the field experiments was the proportion of experimental snails that got infected – that’s a perfectly sensible response variable to look at, but I’m not comfortable with that being called the Force of Infection – “proportional prevalence” (at the group level) or probability of “being infected” (at the individual level) would seem perfectly fine terms. The FOI though is really something a bit different, though related: for an SI-type framework it would be something like $FOI = -\ln(\text{proportion infected})$. I’d suggest either the authors rephrase what they call it on Line 119 or, perhaps preferable to keep the connection with how they subsequently break it down, I would use that transformation in the subsequent model fitting (ie, use $-\ln(\text{proportion infected})$ as the response being fitted to for the equation on Line 133).

- Line 237 – I wonder if rather than using 2 GLMs it would be better to use a single GLM for all the data, with a possible Treatment x Source interaction term?

- Figure 2 – it would be interesting to see a plot of host density x infestation prevalence (effectively panels B x D) to give an indication of how density of infecteds changes across the season, and how this relates to infections in the sentinels. Again, I would suggest relabelling the y-axis of panel A to read “Weekly proportion infected” or similar (or transform the data, as suggested above, to make it more directly related to FOI).

- Figure 3 – would be useful to indicate more clearly on the figure which panels relate to the field experiment and which to the lab. It would also be useful to switch them round, since the field results are discussed first in the text.

Referee: 2

Comments to the Author(s)

Hopkins et al. present an interesting study examining the transmission of a symbiont between two different host species. The authors attempt to tease apart the impact of host density, host infectiousness, and host prevalence on symbiont transmission. The methodologies are interesting and yield some interesting results, but I do have a few concerns regarding data comparisons that are made (more below). The manuscript deals with multiple different species and multiple different experiments that can be hard to keep track of at times. This isn’t surprising given the multiple facets of the study and the authors do a good job of setting up the work (especially in the figures which are pivotal and greatly aid in understanding the study), but I think some work still could go in to improving clarity. This is especially true for the first experiment, which I believe needs more details from the previous work included in this manuscript. While the results are certainly interesting, I think the introduction over sells the importance of the study and tries to hype the study beyond what may be defensible. Primarily, many of the results can fall under the umbrella of host preference, host specificity, and host competency, all of which are fairly well described phenomena which I feel this paper could spend more time exploring in the introduction. The results are most certainly of value and worth seeing in print, here or elsewhere, with some fine tuning and careful consideration on revisions. I break down my specific points section by section below, highlighting major and minor points within each section.

Title: I would cut ‘strong’. How is strong being quantified? Is it relative to something else? I would also replace superspreader. The word appears little in the actual text and tends to be more a trait of an individual within a species than of a species itself in epidemiological contexts. It seems like host competence or infectiousness would be more specific for this context.

Keywords:

I think some more could be added here. Maybe: host-choice, specificity, species interactions, force of infection... I think you can hit on more elements.

Abstract:

Good

Introduction:

The introduction is overall well written and makes many good points. My main issue is that the points discussed in the introduction feel somewhat disjunct from the discussion and focus largely on the aspects of the first experiment and less so on the other two. I feel host preference and specificity warrant more introduction.

Methods:

My major concerns arise largely from the methods, primarily that I feel some controls may be missing that could provide valuable information.

Experiment 1: Much of this data appears to be coming from a previous experiment (which is fine), but we need some more background on the connection between the previous and current work. Were they done in the same year? If not, what issues arise from comparing these data? Were snail and symbiont densities similar in both years? How was sampling done? etc. Also, were lab experiments done with caged snails? If not, comparing any lab and field results may not be valid. What were cages constructed from? How could this influence snail interactions and symbiont transmission? I am sure much of this is discussed in the other paper, but some of these details seem pertinent (even if in a supplement).

I enjoyed the implementation of the equations. They work out quite intuitively. For readers not mathematically inclined I think a really brief mention on the shape of these curves (mostly asymptotic) would be good. Additionally, mentioning how your experiment generates the parameter estimates (which were measured and which were estimated via Bayesian inference) would be good.

Line 150-151: Is there a specific reason these ranges were chosen?

Experiment 2: Does infestation impact contact rates? If high loads of symbionts alter host mobility, how applicable are your results? Or if load is correlated with size how do you separate this collinearity?

Line 164: I am not sure how to interpret the ..., does this mean 1 2 3 5 7 9 or single snail increments until 14? The associated figure appears to show a data point at 15.

Line 171: Why remove trematode infected snails? I know trematodes can have attracting forces on other snails (See papers by Eliuk et al. Canadian Journal of Zoology and by Gray et al. Foci of transmission). That is a good enough reason, but trematodes are common in snails so might warrant a quick mention.

Line 181: How do the symbionts reproduce? Could symbiont eggs come in on periphyton and alter your experiment? Many trematode eggs remain in sediment until ingested also.

Line 197: Is there a reason to take the mean of the beaker instead of using beaker/ trial as a random factor to explain variance in contact while boosting sample size a bit?

Line 209: Since you have this data can't you just look at the distribution instead of assuming its shape?

Experiment 3: The addition of the terms receiver and donor is A LOT to track for the reader. I suggest in the figure axes to include terms like alternative and focal that the reader has adjusted to to help orient the reader. Overall tracking Helisoma, Physa, alternative, focal, donor, receiver, sourced became a bit overwhelming and required a lot of back and forth which made a lot of the manuscript feel disjointed.

In this experiment, the authors use symbionts sourced from different snails. But I am wondering if the environmental conditions at the time of collection couldn't have influenced your results. If the snail hosts were at different densities in the environment at the time of sampling, could this have influenced or habituated the symbiont to a certain stimulus? The authors mention that in a previous experiment the symbiont readily attached to the snails they were presented with, but in this experiment, many didn't attach to a host. This would seem to suggest that something abnormal may have been happening. If Helisoma densities were really high when symbionts were collected, perhaps symbionts were biased to seek out abundant hosts?

Additionally, were symbionts for this experiment randomized? If symbionts from a certain host were used in this experiment couldn't this generate pseudoreplication error associated with relatedness and symbionts originating from the same host?

Experiment 2 and 3: Why were Focal to alternative transmission experiments (and alternative to alternative) not done? These controls would seem to provide valuable data about interspecific and intraspecific transmission that is unaccounted for here.

Host size consistently arises as a factor in many snail-symbiont interactions. How does your experimental design account/ control for host size?

Results:

Line 294: Are there differences in DIC that are standardized for model support?

Figures are nice and very necessary.

Discussion:

Could hosts be showing some sort of frequency dependent host selection based on the environment they were raised in?

Given the limited genetic work on the symbiont, might you be working with multiple cryptic species that have variable host preferences?

Is there a fitness cost of host switching for a symbiont? How do symbionts reproduce, and could this have an effect on preferred paths of transmission?

Figure 1: Give the definitions on the variables in the equation.

Figure 2C: Go ahead and add units to infection intensity for clarity. Also, this data looks more negative binomial than Poisson, or at least Poisson with overdispersion beyond 1. Consider using a different distribution if it is a better fit.

Could the x axis on all figure 2 be made into a calendar date so this data may be more useful for judging other environmental factors? And use by others.

Figure 4: Provide a pseudo-r squared for you curve?

Figure 5: The snail images in the center are really useful, but blend in when looking at the data.

Mention the diagram specifically in the caption or elsewhere because it's super helpful once you figure out what it means.

Some trails appear to have more symbionts than should have been present (10?). Where did the extra symbionts come from?

This is a really interesting system and I will be interested to see more work come out of further studies. Best of luck! And feel free to contact me via the editor if I can be of any assistance.

Author's Response to Decision Letter for (RSPB-2021-1224.R0)

See Appendix A.

RSPB-2022-0084.R0

Review form: Reviewer 1

Recommendation

Accept as is

Scientific importance: Is the manuscript an original and important contribution to its field?

Excellent

General interest: Is the paper of sufficient general interest?

Good

Quality of the paper: Is the overall quality of the paper suitable?

Good

Is the length of the paper justified?

Yes

Should the paper be seen by a specialist statistical reviewer?

No

Do you have any concerns about statistical analyses in this paper? If so, please specify them explicitly in your report.

No

It is a condition of publication that authors make their supporting data, code and materials available - either as supplementary material or hosted in an external repository. Please rate, if applicable, the supporting data on the following criteria.

Is it accessible?

Yes

Is it clear?

Yes

Is it adequate?

Yes

Do you have any ethical concerns with this paper?

No

Comments to the Author

The authors have done a good job of addressing my previous comments, and I have no further comments to make.

Review form: Reviewer 2

Recommendation

Accept with minor revision (please list in comments)

Scientific importance: Is the manuscript an original and important contribution to its field?

Good

General interest: Is the paper of sufficient general interest?

Good

Quality of the paper: Is the overall quality of the paper suitable?

Excellent

Is the length of the paper justified?

Yes

Should the paper be seen by a specialist statistical reviewer?

No

Do you have any concerns about statistical analyses in this paper? If so, please specify them explicitly in your report.

No

It is a condition of publication that authors make their supporting data, code and materials available - either as supplementary material or hosted in an external repository. Please rate, if applicable, the supporting data on the following criteria.

Is it accessible?

N/A

Is it clear?

N/A

Is it adequate?

N/A

Do you have any ethical concerns with this paper?

No

Comments to the Author

Given the available data the authors have done a good job addressing my original comments. The study opens the door to many unanswered questions in the field, but they do not pertain to the goal of the study and will make excellent material for future studies (as is the result of any good study).

A few minor comments below.

Keywords - Does epidemics really fit here? You can argue the results may apply to epidemics, but the study focuses very little on this.

Line 120-121: reword, "... unless hosts are in direct contact."

Line 153: Tell us which two wild hosts. We can assume, but make it clear.

Line 185-186: A 50% burn-in seems like a lot. Not sure if this is standard (or if a standard burn-in really exists for these applications). Likley was necessary as the uninformative priors took a long time to generate stable traces?

Line 210: Perhaps associate this reference with the reference #? Will make it easier for the reader to find the study and cite you!

Line 270: delete 'a few'. This qualifier sort of implies to the reader that the number is insignificant but the reader should make that decision based on their judgement.

Line 283: Give a reference number for this ref again?

Line 348: Should this be figure 3?

Additional thoughts and musings:

Could Chaetogaster show density dependent dispersal? If one donor snail had 50 Chaetogaster, symbionts may be more likely to attempt dispersal than from a donor snail that had 10 symbionts. Not sure if this was discussed or if I just missed it, but seems like it is worth mentioning.

In Line 393-396 the authors discuss the issues associated with symbiont attachment to eventual donor snails. Failure to attach to the donor likely filters out the highly host specific symbionts that WOULD HAVE jumped back to their original host. Wouldn't this filtering step be driving some low interspecific transmission? Could this primary result be an artefact of this experimental design?

I still find figure 5 a bit confusing likely because this particular experiment was complex. Make it clear what the x axis is showing, is it number of symbionts that are left on the donor after contact with the receiver? I had a bit of trouble figuring out what the points meant and how this

generated your statistical results. Is there any way that donor and receiver points could be connected by lines to show how the two counts are related to one another? Might make things busier but could also show the relationship between the set of points.

Best of luck on future projects!

Decision letter (RSPB-2022-0084.R0)

14-Feb-2022

Dear Dr Hopkins

I am pleased to inform you that your manuscript RSPB-2022-0084 entitled "Host preferences inhibit transmission from potential superspreader host species" has been accepted for publication in Proceedings B.

The referee(s) have recommended publication, but also suggest some minor revisions to your manuscript. Therefore, I invite you to respond to the referee(s)' comments and revise your manuscript. Because the schedule for publication is very tight, it is a condition of publication that you submit the revised version of your manuscript within 7 days. If you do not think you will be able to meet this date please let us know.

Sincerely,

Dr Sasha Dall

Associate Editor

Board Member

Comments to Author:

Many thanks for your careful attention to the previous round of reviewer comments. There are just a few outstanding issues that need to be resolved in the next version of the MS (see comments from Reviewer 2, below).

Reviewer(s)' Comments to Author:

Referee: 1

Comments to the Author(s).

The authors have done a good job of addressing my previous comments, and I have no further comments to make.

Referee: 2

Comments to the Author(s).

Given the available data the authors have done a good job addressing my original comments. The study opens the door to many unanswered questions in the field, but they do not pertain to the goal of the study and will make excellent material for future studies (as is the result of any good study).

A few minor comments below.

Keywords - Does epidemics really fit here? You can argue the results may apply to epidemics, but the study focuses very little on this.

Line 120-121: reword, "... unless hosts are in direct contact."

Line 153: Tell us which two wild hosts. We can assume, but make it clear.

Line 185-186: A 50% burn-in seems like a lot. Not sure if this is standard (or if a standard burn-in really exists for these applications). Likley was necessary as the uninformative priors took a long time to generate stable traces?

Line 210: Perhaps associate this reference with the reference #? Will make it easier for the reader to find the study and cite you!

Line 270: delete 'a few'. This qualifier sort of implies to the reader that the number is insignificant but the reader should make that decision based on their judgement.

Line 283: Give a reference number for this ref again?

Line 348: Should this be figure 3?

Additional thoughts and musings:

Could *Chaetogaster* show density dependent dispersal? If one donor snail had 50 *Chaetogaster*, symbionts may be more likely to attempt dispersal than from a donor snail that had 10 symbionts. Not sure if this was discussed or if I just missed it, but seems like it is worth mentioning.

In Line 393-396 the authors discuss the issues associated with symbiont attachment to eventual donor snails. Failure to attach to the donor likely filters out the highly host specific symbionts that WOULD HAVE jumped back to their original host. Wouldn't this filtering step be driving some low interspecific transmission? Could this primary result be an artefact of this experimental design?

I still find figure 5 a bit confusing likely because this particular experiment was complex. Make it clear what the x axis is showing, is it number of symbionts that are left on the donor after contact with the receiver? I had a bit of trouble figuring out what the points meant and how this generated your statistical results. Is there any way that donor and receiver points could be connected by lines to show how the two counts are related to one another? Might make things busier but could also show the relationship between the set of points.

Best of luck on future projects!

Author's Response to Decision Letter for (RSPB-2022-0084.R0)

See Appendix B.

Decision letter (RSPB-2022-0084.R1)

28-Feb-2022

Dear Dr Hopkins

I am pleased to inform you that your manuscript entitled "Host preferences inhibit transmission from potential superspreader host species" has been accepted for publication in Proceedings B.

Data Accessibility section

Open Access

Paper charges

Sincerely,

Dr Sasha Dall

Appendix A

07-Aug-2021

Dear Dr Hopkins:

I am writing to inform you that your manuscript RSPB-2021-1224 entitled "Strong host preferences inhibit transmission from potential superspreader host species" has, in its current form, been rejected for publication in Proceedings B.

This action has been taken on the advice of referees, who have recommended that substantial revisions are necessary. With this in mind we would be happy to consider a resubmission, provided the comments of the referees are fully addressed. However please note that this is not a provisional acceptance.

Sincerely,

Dr Sasha Dall
mailto: proceedingsb@royalsociety.org

Associate Editor
Board Member: 1
Comments to Author:

Both reviewers agree (and I concur) that this is an interesting, important manuscript. The logic is nuanced and yet clearly presented - this is a complicated study, but it is written with such skill that the arguments are quite easy to follow. The MS tackles an extremely timely topic with implications for the transmission of emerging infectious diseases. There is a lot to be excited about in this MS.

We thank you and your reviewers for your thoughtful and thorough review of our manuscript. As we describe below, we have revised our manuscript to address your comments, and we believe that the clarity of our manuscript is much improved as a result.

However, the reviewers have identified several weaknesses that must be addressed if we are to consider a revised version of the MS. In particular, please pay attention to the following:

1. **METHODOLOGICAL DETAILS** - We'll need more detail on how the field measurements of snail density and infection prevalence were conducted. At the moment, it would be impossible to replicate these protocols with the information given in the Methods, and we're unable to evaluate the rigor of the snail survey without more detail on how it was conducted. Similarly for the experiments - the MS is very reliant on reference to previous studies. We'll need enough detail on all protocols for a reader to replicate the study without referring to previously published work.

We agree with this point and have focused our revision on making sure that it contains every detail necessary to replicate our work. We have added the most important details to the main text; you can see these substantial additions in the tracked changes version of the manuscript. In the interest of keeping an already complex and long methods section digestible for readers (and keeping our manuscript within page limits), we added the rest of the details to a new Supplementary Materials document. If you would prefer, we can incorporate the details from the new Supplementary Materials into the main text. Below, we describe exactly how we addressed each of the comments relating to methodological details.

2. **FORCE OF INFECTION** - Please see Reviewer 1's comments regarding the "Force of Infection" term. It may be necessary either to choose different terminology or re-calculate this value.

This is an excellent point, and we are glad that Reviewer 1 caught it. We should not have called the proportion of snails infested by the end of the experiment the "force of infection"; we were using that term too loosely there, because the force of infection only refers to the instantaneous rate. We have reworded the relevant sentence and the axis on Figure 1 accordingly. Note that in some instances, we used "force of infection" correctly (i.e., the force of infection model) and those sections therefore have not been changed; we only incorrectly applied the term to our field data.

3. **FRAMING** - Consider Reviewer 2's comments about framing. A lot of literature

already exists on host preference, host specificity, and host competency, so please spend more time outlining this context in the Introduction.

We have now added an additional paragraph to the introduction describing how symbiont characteristics, like host preference, can affect transmission. We describe this in more detail below.

Reviewer(s)' Comments to Author:

Referee: 1

Comments to the Author(s)

This manuscript presents field and lab experimental results that seek to understand the role of host preference in determining rates of cross-species transmission for multi-host parasites. The ms is well written, with the underlying logic clearly laid out, and the experiments a well thought through, following a logical progression, and are sensibly analysed. Overall the results are clear – despite this symbiont being a host generalist, and able to infest both host species studied, there are clear preferences for the ‘natal’ host, thereby limiting rates of cross-species transmission, even when provided with the opportunity. As such this work provides a neat refinement of our current understanding of multi-host transmission dynamics – such preferences are probably well enough known to be important for vector-borne pathogens, but this work highlights similar host choice effects for parasites with active transmission stages that have the capacity to introduce an element of host choice into the transmission dynamics. I have a few comments that I would like to see the authors consider:

Thank you! We appreciate your careful review and have addressed each of your comments.

- Line 87: “Symbiont transmission occurs predominantly through host contact...” – it might be useful to further clarify/quantify this for those of us who are less familiar with the system. The off-host risks to the parasite are picked up later on in the Discussion, but some clearer statement here would help us understand better the importance of true host-to-host contact, as measured in the subsequent experiments.

Excellent point. We have added a paragraph about *Chaetogaster* reproduction and transmission to the methods section to clarify these points.

- Line 118 – how much are the mesh bags likely to affect transmission? If (as above) transmission really is through contact, could the bags be reducing transmission rates in the field? And is this likely/possible to differ for inter- and intraspecific contact rates?

We do not think that the mesh bags inhibit transmission, and importantly, we cannot envision that the enclosures differentially affect interspecific versus intraspecific contact rates. While we did not quantify contacts in the wild, we observed both wild *Physa* and wild *Helisoma* on the enclosures and contacting the sentinel snails. As we now explain

in the methods and new Supplementary Materials, snails inside enclosures and wild snails could contact through the mesh, and they were often observed doing so. Furthermore, the mesh size is larger than the width of an individual oligochaete, so symbionts could pass through easily, while snails could not. It is possible that transmission rates are lower through the mesh than through an unfettered contact, but this possible effect of the enclosures should not have differed between intra- vs. interspecific contacts.

- In the field experiment the snail community (densities and infection prevalences of both host species) were “surveyed”. I think we need more information here – presumably these data were used to give the IF, NF, IA, NA parameters in the subsequent FOI equation – in which case it is important to know how they were estimated and how intensive the surveying was; estimating the beta and k values may well be quite sensitive to these input parameters, so it would be good to have a better handle on how reliable and accurate they are.

Great point. We now include the details needed to understand or replicate this part of the study either in the main text or the new Supplementary Materials.

- Line 119 – the response variable from the field experiments was the proportion of experimental snails that got infected – that’s a perfectly sensible response variable to look at, but I’m not comfortable with that being called the Force of Infection – “proportional prevalence” (at the group level) or probability of “being infected” (at the individual level) would seem perfectly fine terms. The FOI though is really something a bit different, though related: for an SI-type framework it would be something like $FOI = -\ln(\text{proportion infected})$. I’d suggest either the authors rephrase what they call it on Line 119 or, perhaps preferable to keep the connection with how they subsequently break it down, I would use that transformation in the subsequent model fitting (ie, use $-\ln(\text{proportion infected})$ as the response being fitted to for the equation on Line 133).

We are so glad that the reviewer caught this! We should not have called the proportion of snails infested by the end of the experiment the “force of infection”; we were using that term too loosely. We have reworded the relevant sentence and the axis on Figure 1 accordingly.

- Line 237 – I wonder if rather than using 2 GLMs it would be better to use a single GLM for all the data, with a possible Treatment x Source interaction term?

Good idea! As expected, this does not change our results, but the significant interaction term might help readers interpret the results more easily than comparing two models. We’ve incorporated this change throughout.

- Figure 2 – it would be interesting to see a plot of host density x infestation prevalence (effectively panels B x D) to give an indication of how density of infecteds changes across the season, and how this relates to infections in the sentinels. Again, I would

suggest relabelling the y-axis of panel A to read “Weekly proportion infected” or similar (or transform the data, as suggested above, to make it more directly related to FOI).

We changed the label to read "weekly proportion infested" and we added additional dashed lines in panel B to illustrate the density of infected snails.

- Figure 3 – would be useful to indicate more clearly on the figure which panels relate to the field experiment and which to the lab. It would also be useful to switch them round, since the field results are discussed first in the text.

We made both suggested changes to this figure.

Referee: 2

Comments to the Author(s)

Hopkins et al. present an interesting study examining the transmission of a symbiont between two different host species. The authors attempt to tease apart the impact of host density, host infectiousness, and host prevalence on symbiont transmission. The methodologies are interesting and yield some interesting results, but I do have a few concerns regarding data comparisons that are made (more below). The manuscript deals with multiple different species and multiple different experiments that can be hard to keep track of at times. This isn't surprising given the multiple facets of the study and the authors do a good job of setting up the work (especially in the figures which are pivotal and greatly aid in understanding the study), but I think some work still could go in to improving clarity. This is especially true for the first experiment, which I believe needs more details from the previous work included in this manuscript. While the results are certainly interesting, I think the introduction over sells the importance of the study and tries to hype the study beyond what may be defensible. Primarily, many of the results can fall under the umbrella of host preference, host specificity, and host competency, all of which are fairly well described phenomena which I feel this paper could spend more time exploring in the introduction. The results are most certainly of value and worth seeing in print, here or elsewhere, with some fine tuning and careful consideration on revisions. I break down my specific points section by section below, highlighting major and minor points within each section.

Thank you very much for your careful review of our paper. We explain how we have addressed these points below, and we think the clarity of our paper is much improved as a result.

Title: I would cut 'strong'. How is strong being quantified? Is it relative to something else?

We removed this word from the title.

I would also replace superspreader. The word appears little in the actual text and tends to be more a trait of an individual within a species than of a species itself in

epidemiological contexts. It seems like host competence or infectiousness would be more specific for this context.

We agree that superspreader is usually used in an individual context in epidemiology and disease ecology, but it is also used to refer to superspreader species in the context of community transmission. We decided to keep this term because we are specifically referring to the framework outlined by Fenton et al. (2015), and it would be more confusing to use their framework but also rename the terms.

Fenton A, Daniel G. Streicker, Petchey OL, Pedersen AB. 2015 Are all hosts created equal? Partitioning host species contributions to parasite persistence in multihost communities. *Am. Nat.* **186**, 610–622. (doi:10.1086/683173)

Keywords:

I think some more could be added here. Maybe: host-choice, specificity, species interactions, force of infection... I think you can hit on more elements.

Good point! We thought we were limited to three keywords, but we could actually include six, so we added three more.

Abstract:

Good

Introduction:

The introduction is overall well written and makes many good points. My main issue is that the points discussed in the introduction feel somewhat disjunct from the discussion and focus largely on the aspects of the first experiment and less so on the other two. I feel host preference and specificity warrant more introduction.

We agree; in trying to keep the paper short, we hadn't covered host preference and specificity enough in the introduction. We've now added a relevant paragraph to the introduction.

Methods:

My major concerns arise largely from the methods, primarily that I feel some controls may be missing that could provide valuable information.

Experiment 1: Much of this data appears to be coming from a previous experiment (which is fine), but we need some more background on the connection between the previous and current work. Were they done in the same year? If not, what issues arise from comparing these data? Were snail and symbiont densities similar in both years? How was sampling done? etc. Also, were lab experiments done with caged snails? If not, comparing any lab and field results may not be valid. What were cages constructed from? How could this influence snail interactions and symbiont transmission? I am sure much of this is discussed in the other paper, but some of these details seem pertinent (even if in a supplement).

We agree that we were relying too heavily on readers looking up our previous paper. To improve the readability, we've added the most important missing methodological details to the methods in the main text, and we also added a detailed supplement for the remaining information. Among these edits, we clarify that the previously published study was from the same year (it was the same survey/study, we just did not publish the *Physa* data previously). We also clarify that the field enclosures did not seem to inhibit snail contacts; we often observed enclosure snails and wild snails contacting and even mating through the mesh enclosures.

I enjoyed the implementation of the equations. They work out quite intuitively. For readers not mathematically inclined I think a really brief mention on the shape of these curves (mostly asymptotic) would be good. Additionally, mentioning how your experiment generates the parameter estimates (which were measured and which were estimated via Bayesian inference) would be good.

Thanks! We now more clearly indicate which parameters are estimated from the models and note that these are nonlinear relationships.

Line 150-151: Is there a specific reason these ranges were chosen?

We now use parentheticals to explain why we chose these values; the ranges are meant to be "uninformative" and thus represent the lowest and highest possible values we could expect. In the case of transmission rates, technically a transmission rate of 1000 or 1 million could go in a model, but it would be biologically nonsensical at this scale.

"We fit this model to our paired field enclosure and cross-sectional survey datasets using a Bayesian framework with non-informative uniform priors, where we assumed that transmission rates (β_{FF} and β_{FA}) were between 0 and 2 (with 2 being an extremely high value) and that density-dependence parameters (k_{FF} and k_{FA}) were between 0 and 1 (see above)."

Experiment 2: Does infestation impact contact rates? If high loads of symbionts alter host mobility, how applicable are your results? Or if load is correlated with size how do you separate this collinearity?

To our knowledge, typical infestation intensities do not affect mobility of sexually mature *Helisoma* or *Physa* snails. *Chaetogaster* might affect mobility of newly hatched snails, but we did not use snails that small here. We also removed any snails on the morning of their trial if they appeared lethargic and adjusted the density treatment accordingly (also described in methods). For this particular experiment, wild snails varied in many ways – size, natural *Chaetogaster* infestation loads, potential pre-patent trematode infections that were not detected by shedding cercariae, etc. – and if these variables influence contact rates, it mimics what happens in the wild. We did not try to quantify all of these sources of variability in our models, which were not run at the individual-level. Instead, we tried to eliminate any obvious sources of variability (excluding trematode-shedding

snails, tiny snails), used a large sample size, randomly assigned snails to tanks and trials, and averaged contact rates across the three focal snails per tank to minimize the effect of any individual variation on study outcomes.

Line 164: I am not sure how to interpret the ..., does this mean 1 2 3 5 7 9 or single snail increments until 14? The associated figure appears to show a data point at 15. In this line, we now list every treatment group to avoid confusion. In a few cases, our planned density treatment was reduced because a snail was dead or lethargic on the morning of the trial. We moved this detail up in the methods to improve clarity.

Line 171: Why remove trematode infected snails? I know trematodes can have attracting forces on other snails (See papers by Eliuk et al. Canadian Journal of Zoology and by Gray et al. Foci of transmission). That is a good enough reason, but trematodes are common in snails so might warrant a quick mention.

Good point. We now briefly specify that we removed obviously trematode-infected snails because trematodes may affect snail behavior.

Line 181: How do the symbionts reproduce? Could symbiont eggs come in on periphyton and alter your experiment? Many trematode eggs remain in sediment until ingested also.

The symbionts are annelids, and if they decide to sexually reproduce at the end of fall, their cocoons can end up in sediment (though they have rarely been observed on snails or off snails, and thus we think contamination extremely unlikely). However, this is not an important factor for our study. For the contact rate experiment where we added periphyton to our experimental containers, we used wild-caught snails, many of which had *Chaetogaster* from the wild. Therefore, it should not matter if we accidentally introduced some additional *Chaetogaster* to the arenas. For the field enclosure experiment and the transmission success experiment, we did use lab-reared, uninfested snails that should not have had any *Chaetogaster*. These snails were not fed periphyton collected from ponds, so there was no opportunity for contamination. Additionally, we would routinely check the stock tanks to be sure that the snails did not have *Chaetogaster*, and thus we know that contamination of the stock snails did not occur. We hope that the additional details provided in the methods section addresses this concern.

Line 197: Is there a reason to take the mean of the beaker instead of using beaker/ trial as a random factor to explain variance in contact while boosting sample size a bit?

The two methods (mean response or individual response with random effect) both control for the non-independent observations of snails in the same tank. We opted to use the mean response because it was easier to implement in a nonlinear Bayesian statistical model. Note that the random effects method would have allowed us to quantify variation among individuals, but it would not actually increase our sample size, because the design only allows for 48 independent observations (at the tank level).

Line 209: Since you have this data can't you just look at the distribution instead of assuming its shape?

The shape of the distribution was approximately normally distributed (except that contact rates could be zero-truncated), and that is why we made this assumption in our Bayesian model. To clarify, as in generalized linear models implemented in a frequentist framework, we needed to specify the most appropriate distribution for our response variable, and the normal distribution seemed most appropriate for continuous mean contact rates. As we describe in the methods, model fits and predictions supported that our model provided a good fit to the data.

Experiment 3: The addition of the terms receiver and donor is A LOT to track for the reader. I suggest in the figure axes to include terms like alternative and focal that the reader has adjusted to to help orient the reader. Overall tracking *Helisoma*, *Physa*, alternative, focal, donor, receiver, sourced became a bit overwhelming and required a lot of back and forth which made a lot of the manuscript feel disjointed.

We recognize the difficulty with communicating the receiver/donor ideas and went through several iterations to find language that was as concise and clear as we could make it. We appreciate your help in brainstorming how best to communicate this complicated design. In addition to adding the suggested axis labels, we also added a new diagram to the new Supplementary Materials that further illustrates the experimental design.

In this experiment, the authors use symbionts sourced from different snails. But I am wondering if the environmental conditions at the time of collection couldn't have influenced your results. If the snail hosts were at different densities in the environment at the time of sampling, could this have influenced or habituated the symbiont to a certain stimulus? The authors mention that in a previous experiment the symbiont readily attached to the snails they were presented with, but in this experiment, many didn't attach to a host. This would seem to suggest that something abnormal may have been happening. If *Helisoma* densities were really high when symbionts were collected, perhaps symbionts were biased to seek out abundant hosts?

You are correct that we did the two experiments at different times in the season and we now include those dates in the supplement and explain how they correspond to seasonal snail density trends. We did the alternative-source (*Physa*-source) experiment when *Physa* were abundant (and more abundant than *Helisoma*) and the focal-source (*Helisoma*-source) experiment when *Helisoma* were abundant (and roughly equal in density to *Physa*), because those were the times when we could collect enough symbionts for the experiments. As you mention below, one reasonable hypothesis for host preference would be frequency-dependent selection/preference, where symbionts prefer *Physa* when *Physa* are most abundant and prefer *Helisoma* when *Helisoma* were most abundant. And the results of our transmission success experiment are consistent with this idea. However, this is not a problem or flaw in our experimental design, because it mimics what happens in the wild. Instead, this is a reasonable explanation for why we see host preferences and low interspecific transmission rates. We now cover this in more detail in our discussion.

We think this concern could have stemmed from a sentence that may have miscommunicated some of our previous work. We meant to say that we have only ever done intraspecific transmission experiments among *Helisoma* snails, and in those experiments, transmission rates have always been high; that contrasts the low *interspecific* transmission rates and low symbiont attachment rates observed in this study for the first time. To clarify that point, this sentence now reads: “In contrast to our previous experimental work in this system, which focused only on intraspecific transmission among focal *Helisoma* snails, we recovered many *C. limnaei* from the bottoms of cups that were not attached to snails during the alternative-source experiment.” Note that transmission rates were not low in the treatment group that corresponds to our previous intraspecific work (*Helisoma*-sourced symbionts and *Helisoma*-to-*Helisoma* transmission).

Additionally, were symbionts for this experiment randomized? If symbionts from a certain host were used in this experiment couldn't this generate pseudoreplication error associated with relatedness and symbionts originating from the same host?

We did not randomly assign symbionts to hosts; this would likely be impossible, because the symbionts quickly die or become stuck to debris when removed from the host, so the time required to randomize hundreds of symbionts would likely be too long for a sufficient number to survive and be viable. Instead, we used a haphazard sampling approach: symbionts removed from wild snails were added to a common container and then when there were enough total symbionts for the whole experiment, we haphazardly picked symbionts out of the pooled container and added the correct number of symbionts to each experimental snail. We did not keep track of how many individual wild snails were used to create the symbiont pools, but because most wild snails have 0-5 symbionts and we needed >400 symbionts per experiment, we were removing symbionts from several dozen snails from across the pond. We do expect many, if not all, of those symbionts to be closely related, because the worms reproduce via asexual reproduction, but they were not all collected from just a few snails. Realistically, given the high number of animals in this pooled container and the numerous source hosts, the distribution of symbionts was probably close to random. We've clarified details about *Chaetogaster* reproduction in a new paragraph in the methods and we've added these other details to the new supplemental methods.

Experiment 2 and 3: Why were Focal to alternative transmission experiments (and alternative to alternative) not done? These controls would seem to provide valuable data about interspecific and intraspecific transmission that is unaccounted for here.

We designed the two laboratory experiments to explore mechanisms underlying the results from our field enclosure experiment. Since the field enclosure experiment only used one focal host species, we also only used one focal host species in our laboratory experiments. We agree that it could have been valuable and interesting to repeat the experiments with other host combinations, especially knowing what we know now. But given local resources at the time, it would have been logistically difficult or impossible to add more to what we accomplished here, and the experiments as designed still provide

the necessary information and controls to understand *Helisoma* infection risks, even though we cannot also generalize to *Physa* infection risks.

Host size consistently arises as a factor in many snail-symbiont interactions. How does your experimental design account/ control for host size?

For all experiments, we randomly assigned all snails to treatment groups, such that no treatment groups were likely to have most of the largest or smallest snails. Additionally, we only used “adult” snails or snails big enough to appear sexually mature in this study; we do know from prior work that *Chaetogaster* preferentially disperse to recently hatched snails, but we did not use any snails that small in this study. We did not use snail size as a predictor in our statistical models, because most models did not focus on the individual scale but rather some aggregate (i.e., pond-level prevalence, average contacts per tank). Therefore, any effect of snail size or other individual-level variation would be averaged over.

Results:

Line 294: Are there differences in DIC that are standardized for model support?

Like with AIC, a DIC difference of 2 could be meaningful. However, this question made us realize that our point might have been poorly communicated. We were trying to emphasize that DICs were *not* different and that adding interspecific transmission actually slightly increased DIC, which suggests that this more complicated model was not any better and likely worse. We have revised this sentence to clarify this point. It now reads:

“Correspondingly, adding interspecific transmission to the force of infection model did not improve explanatory power, where the multi-host model that included interspecific (alternative–focal) transmission had a somewhat higher DIC value (pD=2.46, DIC=39.48) than the previously-published single host models that only included intraspecific (focal–focal) transmission (pD=1.98, DIC=38.06)[29].”

Figures are nice and very necessary.

Thanks!

Discussion:

Could hosts be showing some sort of frequency dependent host selection based on the environment they were raised in?

See our response above; we now cover this in our discussion.

Given the limited genetic work on the symbiont, might you be working with multiple cryptic species that have variable host preferences?

It is possible, which is why we discuss the genetics component in our discussion. However, we think it is unlikely, because as we describe in the discussion, *Chaetogaster* removed from one host species can survive and reproduce on a different host species in the lab. Moreover, the one person who has done genetic work on this species hasn't found notable differences among host species. But we do think this is an important area for future research, as we say in the discussion.

Is there a fitness cost of host switching for a symbiont? How do symbionts reproduce, and could this have an effect on preferred paths of transmission?

We have now added an additional paragraph to the methods describing *Chaetogaster* reproduction. As we discuss in the Discussion, it may be that *Chaetogaster* need to preferentially follow cues for their current host species, in case they become dislodged and need to quickly re-find the host. Other than that, there are no known fitness costs for switching. Some host species might be *better* for *Chaetogaster* reproduction; for example, *Helisoma* are bigger than *Physa* and tend to have higher *Chaetogaster* infestation intensities, as we show in Fig 2C. But *Chaetogaster* do not preferentially disperse to *Helisoma* from *Physa*, so whatever benefits are provided by *Helisoma* (if any) might be outweighed by the cost of following cues from multiple host species.

Figure 1: Give the definitions on the variables in the equation.

We added these definitions to the caption.

Figure 2C: Go ahead and add units to infection intensity for clarity. Also, this data looks more negative binomial than Poisson, or at least Poisson with overdispersion beyond 1. Consider using a different distribution if it is a better fit.

Good eye! *Chaetogaster* infection intensity is usually overdispersed relative to a Poisson distribution. In this particular figure, we didn't calculate confidence intervals, nor did we model infection intensity on wild snails as a response variable; it's used as a predictor in the FOI model. We just plotted an average on top of the raw data to aid in visualization of this one predictor variable. We think you might have read the line prior, because we plot mean and 95% Poisson confidence intervals for the snail densities in panel B.

“(B) The wild alternative *P. gyrina* (blue) and focal *H. trivolvis* (orange) snail densities in the pond each week, where the vertical bars are 95% Poisson confidence intervals for each observation. (C) The number of worms per wild snail in the pond each week, with the mean number of worms per snail overlaid in a black-outlined circle.”

Could the x axis on all figure 2 be made into a calendar date so this data may be more useful for judging other environmental factors? And use by others.

We tried this, and it was quite difficult to read because the plot is already packed with information. As a compromise, we added the start and end dates to the figure caption, and the dates of each survey/trial are now included in a table in the new supplement.

Figure 4: Provide a pseudo-r squared for you curve?

Since we are showing the curve on top of the raw data used in the model fitting process, we would prefer not to provide a pseudo- R^2 and instead allow the reader to visualize the fit. This is our preference because methods for pseudo- R^2 s are relatively new for Bayesian models and not yet widely used.

Also see Gelman et al. 2018:

http://www.stat.columbia.edu/~gelman/research/unpublished/bayes_R2.pdf

Figure 5: The snail images in the center are really useful, but blend in when looking at the data. Mention the diagram specifically in the caption or elsewhere because it's super helpful once you figure out what it means.

Good idea! We added a note to the caption emphasizing that the diagram illustrates the panel labels.

Some trails appear to have more symbionts than should have been present (10?). Where did the extra symbionts come from?

Good question! *C. limnaei* can rapidly asexually reproduce, so some experimental units ended up with a few more total symbionts than were initially added. That's why we used proportions of symbionts that dispersed as our response variable, instead of raw counts. We have now explained this in the methods section.

This is a really interesting system and I will be interested to see more work come out of further studies. Best of luck! And feel free to contact me via the editor if I can be of any assistance.

Thank you very much for your thoughtful review! We think the clarity of our paper is greatly improved after getting your insights.

Appendix B

1) A text file of the manuscript (doc, txt, rtf or tex), including the references, tables (including captions) and figure captions. Please remove any tracked changes from the text before submission. PDF files are not an accepted format for the "Main Document".

We have uploaded the main manuscript file as a Word document.

2) A separate electronic file of each figure (tiff, EPS or print-quality PDF preferred). The format should be produced directly from original creation package, or original software format. PowerPoint files are not accepted.

We have uploaded each figure as a separate tiff file.

3) Electronic supplementary material: this should be contained in a separate file and where possible, all ESM should be combined into a single file. All supplementary materials accompanying an accepted article will be treated as in their final form. They will be published alongside the paper on the journal website and posted on the online figshare repository. Files on figshare will be made available approximately one week before the accompanying article so that the supplementary material can be attributed a unique DOI.

We have updated our supplementary material to contain these details and submitted it as a single file.

We have included the following media summary:

Symbionts are organisms that live on a host, such as parasites and beneficial bacteria. This study quantified symbiont transmission among two abundant snail host species that are commonly infested with the same symbiotic worm. The snail species often interact in the wild and frequently contacted in experiments, yet there was almost no measurable symbiont transmission between the snail species in the wild. Symbionts instead strongly preferred their current host species and rarely switched species when given the opportunity. These unexpected results demonstrate that symbionts' preferences can create strong transmission barriers among host species that might otherwise be "superspreader" species.

If you wish to submit your data to Dryad (<http://datadryad.org/>) and have not already done so you can submit your data via this

link [http://datadryad.org/submit?journalID=RSPB&manu=\(Document](http://datadryad.org/submit?journalID=RSPB&manu=(Document) not available) which will take you to your unique entry in the Dryad repository. If you have already submitted your data to dryad you can make any necessary revisions to your dataset by following the above link.

Please see <https://royalsociety.org/journals/ethics-policies/data-sharing-mining/> for more details.

We have uploaded all of our data and scripts to Dryad and updated the data availability statement accordingly.

Hopkins, Skylar; McGregor, Cari; Belden, Lisa; Wojdak, Jeremy (2022), Host preferences inhibit transmission from potential superspreader host species, Dryad, Dataset, <https://doi.org/10.5061/dryad.hmgqnk9jw>

Sincerely,

Dr Sasha Dall

Associate Editor

Board Member

Comments to Author:

Many thanks for your careful attention to the previous round of reviewer comments. There are just a few outstanding issues that need to be resolved in the next version of the MS (see comments from Reviewer 2, below).

Thank you for considering our paper! We have addressed all of the reviewers' comments and respond to each one below.

Reviewer(s)' Comments to Author:

Referee: 1

Comments to the Author(s).

The authors have done a good job of addressing my previous comments, and I have no further comments to make.

Thank you!

Referee: 2

Comments to the Author(s).

Given the available data the authors have done a good job addressing my original comments. The study opens the door to many unanswered questions in the field, but they do not pertain to the goal of the study and will make excellent material for future studies (as is the result of any good study).

Thank you!

A few minor comments below.

Keywords - Does epidemics really fit here? You can argue the results may apply to epidemics, but the study focuses very little on this.

We removed epidemic from the keyword list.

Line 120-121: reword, "... unless hosts are in direct contact."

We made this change.

Line 153: Tell us which two wild hosts. We can assume, but make it clear.

We added the two Latin names in parenthesis after "two wild hosts".

Line 185-186: A 50% burn-in seems like a lot. Not sure if this is standard (or if a standard burn-in really exists for these applications). Likley was necessary as the uninformative priors took a long time to generate stable traces?

There are some rules of thumb for burn-in lengths, but no standard; the burn-in length should be as long as it needs to be to only sample where the model has converged. We used a longer burn-in for our field data to ensure convergence and then we just kept the methods the same for the experimental data (which converged more quickly) to avoid confusing readers with two different burn-in periods. As long as convergence is reached (which we confirmed for both analyses using R_{hat} values and plots of the chains), the longer burn-in period doesn't affect the results.

Line 210: Perhaps associate this reference with the reference #? Will make it easier for the reader to find the study and cite you!

Good catch, thanks! The citation manager seems to have malfunctioned and given us a different parenthetical citation format there. We've switched it to the numerical format.

Line 270: delete 'a few'. This qualifier sort of implies to the reader that the number is insignificant but the reader should make that decision based on their judgement.

We deleted "a few".

Line 283: Give a reference number for this ref again?

Good catch, thanks! We fixed the formatting here.

Line 348: Should this be figure 3?

Yes, thanks! We corrected this and checked the figure designations throughout to make sure that there were not any other errors.

Additional thoughts and musings:

Could *Chaetogaster* show density dependent dispersal? If one donor snail had 50 *Chaetogaster*, symbionts may be more likely to attempt dispersal than from a donor snail that had 10 symbionts. Not sure if this was discussed or if I just missed it, but seems like it is worth mentioning.

We've added a sentence to this in the supplement, where we justify treating infestation as presence/absence (instead of infestation intensity) in the field analysis. We have experimentally quantified *Chaetogaster* transmission in the lab before, and at the infestation intensities that we typically observe in the field, there is no evidence for density-dependent dispersal.

In Line 393-396 the authors discuss the issues associated with symbiont attachment to

eventual donor snails. Failure to attach to the donor likely filters out the highly host specific symbionts that WOULD HAVE jumped back to their original host. Wouldn't this filtering step be driving some low interspecific transmission? Could this primary result be an artefact of this experimental design?

If we understand this question correctly, we actually see the reverse: transmission rates were highest when the receiver snail was the same species as the source snail species from the wild, which we do think is because the symbionts are quickly returning towards the familiar host species. We think the problem mentioned here would be an issue if we only looked at the number of *Chaetogaster* that dispersed. Instead, our statistical models consider how the *proportion* of *Chaetogaster* on the receiver snail varied with treatment group.

I still find figure 5 a bit confusing likely because this particular experiment was complex. Make it clear what the x axis is showing, is it number of symbionts that are left on the donor after contact with the receiver? I had a bit of trouble figuring out what the points meant and how this generated your statistical results. Is there any way that donor and receiver points could be connected by lines to show how the two counts are related to one another? Might make things busier but could also show the relationship between the set of points.

We changed the first line of this caption to say, "Symbiont transmission success rates, where individual points show the number of symbionts remaining on the donor snail versus the number that dispersed to the receiver snail by the end of the experiment."

Best of luck on future projects!

Thank you for your thoughtful review! Our paper has improved greatly from your feedback.